# Landscape Values in a Marina in Granada (Spain): Enhancing Landscape Management through Public Participation

**Ricardo Martín** [1,*] **and Víctor Yepes** [2]

1    Department of Civil Engineering, Universitat Politècnica de València, 46022 Valencia, Spain
2    ICITECH, Department of Civil Engineering, Universitat Politècnica de València, 46022 Valencia, Spain
*    Correspondence: rimarpo@doctor.upv.es

**Abstract:** Landscape values are related to the attributes that people assign to a perceived landscape. They reflect marina user perceptions, thus representing a feedback tool for marina managers to use to verify the degree of user satisfaction. This study focused on identifying and assessing a marina's landscape values. We took Marina del Este (Granada, Spain) as a case study. We considered interviews and a questionnaire to devise methods to enhance the participation of stakeholders and users. First, the SWOT analysis from stakeholder interviews enabled us to collect management's perceptions. Second, the survey gathered the marina's landscape values, comprising 104 respondents from visitors and users. ANOVA and PCA methods were applied to check the suitability of the values. The results showed that the marina should be in keeping with an atmosphere of tranquility and well-being. Nevertheless, there was a need to improve values related to nautical tourism, such as hospitality and maintenance, dealing with the lack of space and an excess of urbanization in the surroundings. Marina managers should consider these outcomes and analyze the points of improvement to establish the causes of these disagreements and propose solutions for the established management model. The perception of stakeholders and users can enable more consensual policies with greater levels of acceptance and involvement.

**Keywords:** landscape value; marina; management; SWOT; ANOVA; PCA

## 1. Introduction

Management is responsible for controlling an organization and dealing with it carefully. This study focuses on identifying and assessing landscape values in a marina through public participation. It also represents an opportunity to enhance public participation in the marina's landscape management solutions.

Nowadays, all marinas essentially offer similar services and amenities for boats. Marina managers seek differentiation formulas to generate a competitive advantage and provide a recognizable image [1]. Some scholars highlight the hospitality business perspective, emphasising the need to offer an attractive atmosphere, better services, higher quality, more personalized care, greater convenience, and greater social and environmental responsibility [2–5]. Others focus their differentiation on a sustainable perspective. These base recommendations on the local community's participation, applying principles of blending into the environment and seeking synergies with the guest experience. It is a research hotspot because environmental aspects, sustainable development, and economic development enhance the social dimension of the marinas [6–8]. It drives the introduction of new services grounded on local natural and spatial potential and the imperative to meet the broader needs of the surroundings [2,3,9–11]. Most of the nautical facilities have approaches that are located between both positions.

The landscape within marinas represents an opportunity to improve their management. A characteristic landscape may enhance the tourist appeal and improve business by achieving a defined image and gaining economic benefits [12,13]. However, there is a lack

of study of landscapes within the management of marinas [14]. Martín and Yepes [1] study the elements that constitute the landscape within marinas. The relevant items to consider in the management of marinas are identified from a landscape perspective [14]. Nevertheless, there is no feedback from users about how the decisions on landscape affect their interests.

The landscape depends on the meaning individuals and society provides [15]. Recognition of the values that people share and attribute to a landscape is essential to landscape identification [16,17]. Landscape values can be defined as attributes people assign to the perceived landscape due to subjective interpretation resulting from the interaction between humans and their environment [18,19]. Therefore, the main target for marina managers is to establish what values people attribute to the landscape of marinas. It is an indirect reflection of past experiences and expectations for the future of users and visitors.

## 1.1. Landscape Values

The original typology of landscape values was developed by Brown and Reed [20], who established a set of 13 values (aesthetic, recreation, biodiversity, life-supporting, economic, learning, historical, cultural, future, intrinsic, spiritual, therapeutic, subsistence) as part of a forest planning process. This typology has been adapted and used for different applications, such as public lands [21], country management [22–24], urban areas [25–28], rural landscapes [29,30], and coastal landscapes [31,32]. However, there is not a one-to-one correspondence when dealing with places and values [26].

Regarding marinas, some scholarships analyze users' perceptions as valuable tools in verifying users' satisfaction levels. Yachters' perceptions are used to identify market segmentation [4] or investigate their quality perceptions' effects on their satisfaction with marina services [5]. However, there needs to be more work done in assessing perceptions and values of the landscape in marinas.

## 1.2. Methodology of Investigation

Public participatory geographic information systems (PPGIS) methodologies are widely used by researchers in examining landscape values. These methods collect and analyze spatial data from places and values in the geographical distribution to provide decision support in planning processes [21]. PPGIS surveys ask users to identify and mark locations on a map and select values based on participants' experiential knowledge according to a typology given [22,28,30,32–34]. However, other methodologies of determining landscape values ask about the values transmitted in pre-identified located points on a territory [26,31,35–37].

Different methodologies are used to examine the differences between outcomes. The analysis of variance (ANOVA) method is used to examine the mean difference [38], while Chi-square tests are applied to explore statistical differences between dependent and independent variables [35]. The correlation coefficients establish the grade of linkage between pairs of variables [26]. Principal component analysis is used to perform factor analysis from the extracted principal component factors [39,40].

## 1.3. Study Objectives

This study seeks to apply the landscape value framework to the evaluation of a marina. The limited landscape research in marinas is why the landscape values in these maritime facilities have yet to be studied. The study was carried out through public participation, which provides outcomes from a theoretical framework to improve marinas' landscape management. Public participation should include experts and people, but it also must recognize the different stakeholders and social groups [41,42]. Attending to the work of Eitier and Vik [43] and translated to marinas, management implies a mutual commitment between parties: managers, stakeholders, and users. Moreover, managers should enhance the participation of the public and other relevant stakeholders in the landscape policies [44]. We addressed the following questions:

RQ1. Which are the landscape values identified, and how did users perceive the marina landscape?

RQ2. How is it possible to verify the suitability of the chosen values?

RQ3. How could public participation in determining landscape values be increased?

Knowing the links between people and the marina is the primary goal of marina managers. Landscape values reflect the marina user perceptions. They represent a feedback tool through which marina managers can verify whether the policies implemented have a satisfactory response from users. They also represent a set of expectations to satisfy. Marina managers must learn from customers' needs and anticipate future expectations [14]. hey also verify whether the developed strategic planning aligns with what the stakeholders perceive appropriate.

## 2. Materials and Methods

### 2.1. Study Loction

We considered Marina del Este, found on the Southern Spanish Mediterranean coast as case study. Specifically, it is located in the town of La Herradura, belonging to the municipality of Almuñecar (Granada). It is the last town on the coast of Granada before entering Malaga (Figure 1).

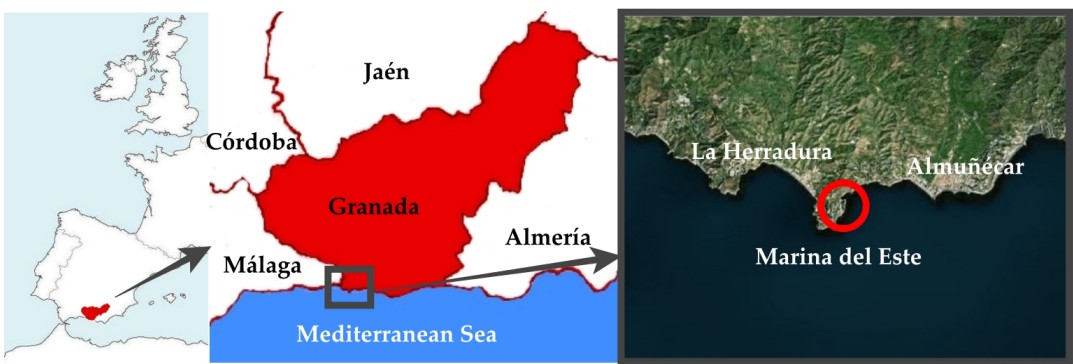

**Figure 1.** Location of Marina del Este.

Marina del Este is sheltered in the countryside, between mountains and the sea. It is close to Punta de la Mona, which protects the marina from strong winds from the west and allows beautiful sights to the east. It was constructed around the Peñón de las Caballas in an original design that combines the harmony of the docks with the rocky whims of nature (Figure 2). It is near magnificent beaches and hidden coves in Maro-Cerro Gordo cliffs. There is not direct access from the land via the A-7 motorway. It is reached by the N-340 road, on the strip between La Herradura and Almuñécar. The final stretch is narrow and winding. For sea access, the reference is a red-light beacon (36°43.39′ N; 3°43.35′ W). The marina has 227 docks, with a maximum length of 35 m. The maximum draft is 6.5 m, being 3.7 in the mouth. With a total surface of about 2 Ha, 1.2 correspond to the inland area.

Marina del Este is very close to protected natural areas, such as the Maro-Cerro Gordo (ES6170002) cliffs and Punta de la Mona cliffs and seabeds (ES6140016). Moreover, they are integrated into the Natura 2000 network as a Specially Protected Area and a Special Protection Area under the Birds Directive. In addition, the Maro-Cerro Gordo cliffs also have the status of the UN's Specially Protected Areas of Mediterranean importance. This proximity highlights the quality of its waters and seabed, making this area a privilege for lovers of diving and scuba diving. However, terrestrial and marine biota also represent a competitive advantage for natural areas' visitors. However, the area near the port is considered developable for urbanization. Currently, intensive construction is being built that, in addition to creating an artificial background, is antagonistic to the natural environment. This proximity to protected natural areas makes the marina environmentally significant. It is reflected in the area winning the Blue Flag award for Marinas over several

years. This award recognizes the environmental quality in marinas that make a special effort in terms of local environmental management and nature. Obtaining the Blue Flag implies that the marina must meet various requirements related to environmental education and information, environmental management, services, safety, and water quality. All this provides the visitor with a reliable guarantee regarding the environmental quality of the marina [45].

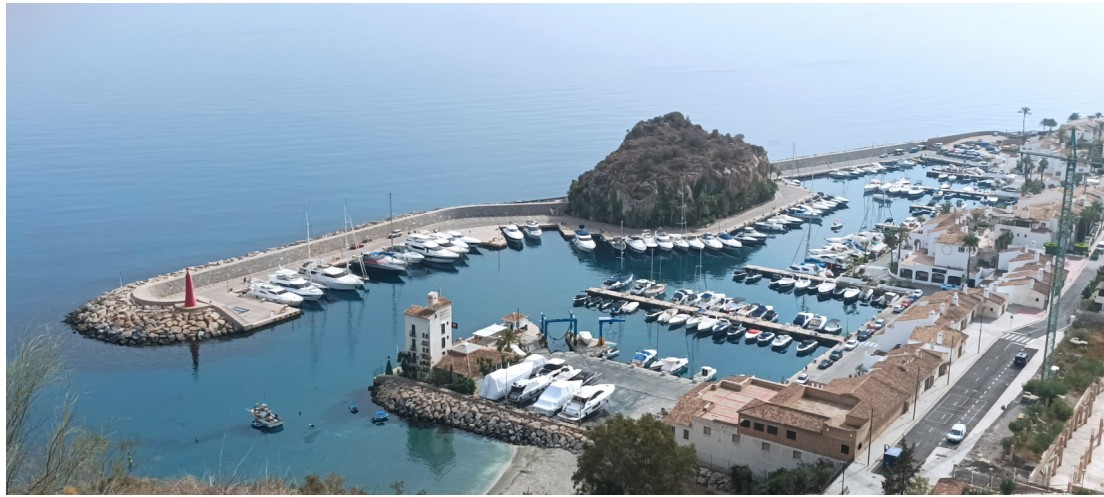

**Figure 2.** General view of Marina del Este with Peñón de las Caballas as a structuring element (photograph by authors, October 2022).

*2.2. Data Collection*

The spatial scale of landscape values studies usually covers large areas or territories, both on regional [28,34,36] and urban [26,46,47] scales. Concerning the case of study, the dimensions of the marina adopted mean this installation was considered a single point when establishing marina landscape values.

Regarding the landscape values to consider, we focused on elements of marinas [12] and research grounded on previous interviews with stakeholders to collecting their opinions and perceptions of the marina. The interviews are qualitative and aimed at collecting local opinions on shaping the landscape through emotion and local experience [48]. They also represent a source of first-line and first-hand knowledge because the attachment to a place depends on a personal connection, which can only be fully understood with actual interaction. Considering the implication of different parties, the consensus should be achieved through dialogue and commitment between all parties to promote the general interest. Nevertheless, this is more an exception than a rule [42]. We attempted to embrace those different sensibilities and standpoints but did so independently. We enacted the following steps to address the issues that arose:

1. Interviews with principal stakeholders;
2. A general questionnaire for marina users.

2.2.1. Interviews

The first step was to identify a set of complementary information from interviews that could be used to obtain a SWOT analysis. This approach related a twofold objective: firstly, it could be compared with that performed by marina managers, and also related management items to landscape {14}; secondly, it served as a basis for establishing the features of the marina and thus determining the nature of the values on which to question users.

In this sense, we identified the different uses allowed within the port concerning Spanish Act 02/2011. It identifies nautical, fishing, and commercial uses. Nevertheless, complementary or auxiliary uses of the above and uses related to port-city interaction are

also considered. We attempted to include participants related to the above uses. Due to constraints on time, we considered the representative persons of the existing uses in the marina: yacht club, commercial and nautical services. Table 1 shows the participants and their affiliations. In order to improve the quality of the responses, we exclusively selected respondents in the position of running a business.

**Table 1.** Participants and affiliations considered.

| Groups | Description | Number |
|---|---|---|
| Yacht club | Maintenance and nautical services | 1 |
| Commercial | Supermarket | 1 |
|  | Real estate business | 1 |
| Nautical | Sailing instructor and nautical consultant | 1 |
|  | Sale, rental, and repair of boats and engines | 1 |
| Total |  | 5 |

We used one-to-one semi-structured interviews, which allowed the interviewees to be free to explain within flexible conduction and tailored to them and their responses to previous questions, taking into account the general aims of the research. In this sense, the interview was prepared prior, considering questions grouped into three blocks (see Appendix A). The first one dealt with the marina's management. The interviewees were asked their opinion on the marina's management, identifying strengths and weaknesses. They were also proposed to provide solutions. The second block examined the idea of landscape. The participants were asked what they understood as landscape and if they knew ELC. Later, they were asked how they should improve the landscape within the marina, taking into account the previous ideas carried out in the above topic. The last block looked into public participation by asking in which way and to what extent the public should participate in the decision-making process.

The interviews were carried out in February 2020 and March 2021 due to the COVID-19 pandemic and the restrictions imposed. They lasted from 30 min to $1^{1/2}$ h. All of them were conducted in the workplaces of the interviewees. The management of the interviews was challenging due to the attitude of some interviewees, who were passionate and willing to complain, sometimes beyond the scope of this research. It required constant control and focusing on the issues. The interviews were not recorded, but notes were written during the conversation. As a result of the interviews with stakeholders, which covered different issues related to management as the first topic, a SWOT analysis was developed. It summarized the way the stakeholders perceive the marina management and, thus, basing how to face the landscape.

### 2.2.2. Questionnaire

After the SWOT analysis, a participatory questionnaire was developed to ascertain users' values assigned to the marina. To this, we the original typologies of landscape values adapted [20], taking into account the following: (1) the elements that make up the landscape in marinas [1]; (2) the elements of marina management that are representative from the point of view of the landscape [14]; and (3) the information obtained from the interviews.

The questionnaire consisted of three parts. In the first section, the person was characterized, including sociodemographic questions, such as gender, nationality, age range, level of education, and occupation. The type of link with the port was established in the second block. In this case, questions about the frequency, seasonality, and reason for visiting the marina were asked. Users also asked about the form of access and the place of origin. Finally, respondents were asked about the values the marina transmitted to them. The attributes to be identified by participants included ten landscape value categories (aesthetic, economic, recreation, environmental quality, learning, social, therapeutic, intrinsic, future, and cultural). Some value categories could be broken down into different single items, giving a total of 27 values (Table A2). For each value, we formulated statements to

assist in understanding it. To deepen the importance given to the set of values, a 5-point Likert-scale assessment was carried out. The respondents were asked to choose between 1 to 5 («Strongly disagree», «Disagree», «Neutral», «Agree», «Strongly agree»). In addition, the survey included open questions. Firstly, respondents were asked what they enjoyed best and least about the marina. Secondly, they were asked to express which two or three images of the marina were most significant to them. Finally, participants were asked to assess the importance of the landscape in the marina, on a scale between 1 and 10.

The survey was intended to cover all users of the marina. To attempt this, a link to the survey website in Spanish and English was sent to frequent users (boaters and visitors) and nearby residents by the stakeholders who had been interviewed. The answer by respondents was anonymous. The survey was open from 20th September until 10th November 2022.

### 2.3. Data Analysis Procedure

The suitability of the values was analyzed using two main methods. First, the reliability of the outcomes was determined through Cronbach's alpha test. Second, the ANOVA method was applied to check whether it was appropriate to group the items in the categories made, considering the value of the category as the average value of the topics that integrate it. Finally, the PCA method was used to evaluate which items could be narrowed down, as well as to identify the most influential factors. For data processing and analysis, we used SPSS©.

Reliability is related to the accuracy or consistency of the measurement. Cronbach's alpha coefficient is a general formula for estimating the reliability in which the response to the items is dichotomous or has more than two values [38–40]. A higher value is interpreted to indicate good internal consistency. A value between 0.70 and 0.95 is considered representative of adequate reliability. We used Cronbach's alpha as a model of internal consistency based on the average inter-item correlation.

The analysis of variance (ANOVA) computes the means of the values and compares the variance of these means against the mean within-group variance. ANOVA was applied to test the null hypothesis, samples came from subpopulations where the mean of the dependent variable was the same as the independent. That is, there were no significant differences between the means observed. The differences were due to chance; therefore, the samples came from the same population. However, the F test displayed a repeated measures analysis-of-variance table. The higher the F value and the lower its significance, the more likely there would be significant differences between the groups considered.

Consequently, if the *p* value associated with the statistic was less than the significance level (0.05), the hypothesis of equality of means was rejected. ANOVA was used in a twofold approach. On the one hand, it was utilized as an indicator of the internal consistency of a group formed by several values. On the other hand, it was deployed to analyze the behavior of a dependent variable in the subpopulations or groups established by the values of a single independent variable (or factor). In this sense, the analysis of the values adopted depends on sociodemographic attributes. We considered gender, age, educational level, frequency and seasonally of visits, as well as origin.

Principal component analysis (PCA) is a data reduction technique used to find homogeneous groups of variables from a large set, considering the correlation between them. Its purpose is to find the minimum number of dimensions capable of explaining the maximum amount of information contained in the data without losing important information. The PCA, therefore, introduces a reduction in the dimensionality of the data. Its advantage lies in simplifying complex problems, reducing the dimensions of multiple impact factors involved in the problem, and driving a better explanation of the correlation between key variables. The PCA was used to analyze the landscape values and, hence, to establish whether the initially determined grouping was adequate. This method could be suitable if the overall Kaiser–Meuer–Okling (KMO) value obtained was greater than 0.7. Additionally, if the results of Bartlett sphericity test were significantly less than 0.05, this indicated a correlation between each item and could be used for PCA.

## 3. Results

### 3.1. Interviews

Related to interviews, opinions. and judgments were obtained depending on the topic, as expected. Similarities were sought to enhance knowledge of the shared ideas, and dissimilarities were analyzed to understand the different standpoints. The results of the interviews were generally conclusive.

Considering the first part of the interview, all participants agreed on the privileged location of the marina, the lack of space, the excess of urbanization in the surroundings, and the lack of coordination between agents related to management. Nevertheless, there was no consensus on the exclusive or open character of the marina and, therefore, whether to improve communication between areas, thus allowing more people to come. It was a familiar sentiment that existing access should be improved but not expanded. The results of Marina del Este's SWOT analysis are shown in Table 2.

**Table 2.** SWOT analysis for Marina del Este from stakeholders' interviews.

| Strengths | Weakness |
|---|---|
| Professional and friendly staff, who convey trust and confidence. | Highest rates in the whole area. |
| Diving centre of reference in the area. | Poor land transport connections and no nearby urban centers. |
| Holding championships, tournaments, and fairs. | A small port with limited berths and no possibility of long promenades. |
| Environmental management procedures. | The existence of disputed areas prevents their exploitation. |
| Possibility of creating a pedestrian promenade with a nearby city center. | Lack of parking space, with congestion problems in the summertime. |
| Possibility of making development space available when outstanding disputes are cleared up. | Lack of parking space, with congestion problems in the summertime. |
| | Excessive urbanization which is growing faster than port services. |
| | Poor land transport links and no nearby urban centres. |
| | Lack of coordination between the marina and the other administrations. |

| Opportunities | Threats |
|---|---|
| High berth demand. | Environmental protection constraints on the development of nautical activities. |
| Idyllic setting, with several protected natural areas nearby. | High seasonality of the tourism product, based on the binomial sun beach. |
| Potential for development of nautical and marine-related activities. | Little nautical tradition, with no strategy to promote it among the population. |
| Good climate with mild winter temperatures and low rainfall. | High exposure to easterly storms. |
| | The existence of unregulated anchorages in the vicinity. |

Regarding the second topic, the landscape is understood in its perception sense given by the various English dictionaries [49–52]: it refers to the portion of territory viewed at a time from one place. In this appearance domain, there is a consensus on the privilege of the environment and scenic landmarks. However, there is a feeling that the marina should be in keeping with an atmosphere of tranquility and well-being. These comments reflect ignorance about the existence of the ELC. Although the visual part of the landscape remains valid, the subjective component was assumed, even if its inclusion within the concept of landscape is unknown. This implicit knowledge facilitates the subsequent understanding of the definition of landscape by the ELC.

Finally, regarding forms of participation, in both cases, stakeholders would like their opinion to be considered. However, for some participants, it needed to be clarified how this participation should be carried out. A second group defines it more precisely: based on prior consultations, the greater transmission of information, and consultations at significant milestones in the process.

### 3.2. Survey

A total of 104 responses were received. The precision level depends on the risk a researcher is willing to take. Using a level of confidence of 95% and considering an unlimited population, the margin of error in the proportion is set at 0.1, which is acceptable

in perception surveys [35]. Despite the efforts taken, including the diffusion among the users (made by the managers of the marina), as well as the nearby population, and directly to visitors at the marina, we have not been possible to maximize the number of responses. The reasons for low participation can differ, ranging from simple apathy to disinterest in the subject being addressed. The latter is evidence of the lack of treatment of this issue in marinas and broader society's ignorance of landscape management. While the total number of responses is sufficient to explore a range of existing values, the relatively low response rate relative to the total potential population means that generalizations about the broader community cannot be made from the results of this study.

The sociodemographic characterization of the responders (Table 3) corresponded primarily to a Spanish (95.2%) male (60.6%) between 45 and 65 years (35.6%), and more highly educated (42.3%), who works as a senior technician, scientist or intellectual (31.7%). The average respondent often (25.0%) visited the marina throughout the year (54.8%) due to sightseeing or sporting activities (54.9%), and access to the port by motor vehicle (70.2%) from the same municipality (43.3%). On the other hand, the fact that there was an even distribution of respondents from youngest to retired people is essential, as previous research on coastal values has been biased toward older respondents [53].

**Table 3.** Characterization of the sample size (N = 104).

| | Sociodemographic Profile | N | % | Link to the Marina | | N | % |
|---|---|---|---|---|---|---|---|
| **Gender** | | | | **Frequency** | | | |
| | Male | 63 | 60.6 | | Sporadically | 21 | 20.2 |
| | Female | 41 | 39.4 | | Sometimes | 20 | 19.2 |
| **Nationality** | | | | | Usually | 26 | 25.0 |
| | Spanish | 99 | 95.2 | | Often | 18 | 17.3 |
| | Other | 5 | 4.8 | | Very often | 19 | 18.3 |
| **Age** | | | | **Seasonality** | | | |
| | >18 | 2 | 1.9 | | Throughout the year | 57 | 54.8 |
| | 28–25 | 7 | 6.7 | | Holydays | 38 | 36.5 |
| | 25–45 | 34 | 32.7 | | Weekends | 9 | 8.7 |
| | 45–65 | 37 | 35.6 | **Reason *** | | | |
| | >65 | 24 | 23.1 | | Work | 14 | 133.5 |
| **Level of education** | | | | | Sailing, and other nautical activities | 30 | 28.8 |
| | No education | 0 | 0.0 | | Catering and leisure | 32 | 30.8 |
| | Primary | 8 | 7.7 | | Sightseeing or sporting activities | 54 | 51.2 |
| | Baccalaureate, intermediate vocational training | 13 | 12.5 | | Other | 10 | 9.6 |
| | Degree, higher vocational training | 44 | 42.3 | **Port access *** | | | |
| | Master's degree, PhD | 39 | 37.5 | | By sea | 20 | 19.2 |
| **Occupational level** | | | | | By motor vehicle | 73 | 70.2 |
| | Unemployed | 9 | 8.7 | | By bicycle | 9 | 8.7 |
| | Unskilled worker | 2 | 1.9 | | On foot | 33 | 31.7 |
| | Skilled worker in agriculture, fisheries, and manufacturing industry, craftsmen | 3 | 2.9 | | Other | 1 | 1.0 |
| | Clerical worker, service worker | 24 | 23.1 | **Origin** | | | |
| | Junior technician, small businessman | 13 | 12.5 | | From the same municipality | 45 | 43.3 |
| | Senior technician, scientist, intellectual | 33 | 31.7 | | From a nearby municipality (<100 km) | 41 | 38.5 |
| | CEO | 12 | 11.5 | | From a distant municipality (>100 km) | 16 | 15.4 |
| | Retired | 24 | 23.1 | | From abroad | 3 | 1.9 |

\* Answers with multiple options.

The Cronbach's alpha coefficient was tested to evaluate the internal consistency of the questionnaire. The obtained value was 0.941, which can be considered to indicate excellent reliability. In addition, the statistics were analyzed in case any element needing

assessment had been suppressed. In no case did the suppression of any items increase reliability significantly. Therefore, all values considered provide internal consistency to the survey. The questionnaire's Kaiser–Mayer–Olkin (KMO) was 0.858, significantly greater than 0.7. The Bartlett sphericity found that the significance was 0.000. Thus, it could be used the PCA.

　　　The basic statistics (mean, standard deviation, standard error or the minimum, maximum, and the upper and lower 95% values of the mean) of each of the 27 landscape values considered for the marina landscape are shown in Tables 4 and A3. The values that were agreed more strongly are as follows (Figure 3): issues related to environmental quality—"Air quality" (4.38), "Natural diversity" (4.17), and "Water quality" (4.11) "Scuba diving" (4.20), "Beauty" (4.17), "Safety" (4.15), "Walking/running" (4.08), "Quiet" (4.07), "Learning of nautical activities" (4.06), and "Yachting" (4.00). However, the values which respondents rated at the maximum value (5) at least half of the respondents were scuba diving (61.5%) and air quality (51.0%).

**Table 4.** Descriptive statistics for landscape values.

| Landscape Values | | Mean | SD | Group Analysis | | |
|---|---|---|---|---|---|---|
| Categories | Items | | | $\alpha$ | F | Sig. |
| **Aesthetic** | | 3.92 | | 0.836 | 14.436 | 0.000 |
| | Beauty | 4.17 | 0.864 | | | |
| | Scenery | 3.71 | 1.183 | | | |
| | Visual compatibility | 3.88 | 1.049 | | | |
| **Economy** | | 3.76 | 0.970 | | | |
| **Recreation** | | 3.86 | | 0.649 | 25.568 | 0.000 |
| | Yachting | 4.00 | 0.975 | | | |
| | Scuba diving | 4.20 | 1.194 | | | |
| | Fishing | 3.27 | 1.310 | | | |
| | Walking/running | 4.08 | 1.049 | | | |
| **Environmental quality** | | 4.22 | | 0.802 | 5.260 | 0.000 |
| | Water quality | 4.13 | 1.011 | | | |
| | Air quality | 4.38 | 0.778 | | | |
| | Natural diversity | 4.17 | 0.908 | | | |
| **Learning** | | 3.93 | | 0.689 | 5.470 | 0.000 |
| | Nautical activities | 4.06 | 1.087 | | | |
| | Natural spaces | 3.80 | 1.234 | | | |
| **Social** | | 3.37 | | 0.860 | 35.789 | 0.000 |
| | Meeting point | 3.75 | 1.134 | | | |
| | Catering | 3.56 | 1.205 | | | |
| | Shopping | 2.71 | 1.196 | | | |
| | Leisure | 3.11 | 1.131 | | | |
| | Exclusivity | 3.72 | 1.038 | | | |
| **Therapeutic** | | 3.97 | | 0.858 | 8.829 | 0.000 |
| | Quietness | 4.07 | 0.873 | | | |
| | Hospitality | 3.88 | 0.952 | | | |
| | Confidence | 3.80 | 0.907 | | | |
| | Safety | 4.15 | 0.856 | | | |
| **Intrinsic** | | 4.00 | 0.985 | | | |
| **Future** | | 3.73 | | 0.471 | 12.110 | 0.000 |
| | Opportunity | 3.93 | 1.007 | | | |
| | Maintenance | 3.52 | 1.052 | | | |
| **Cultural** | | 3.28 | | 0.758 | 4.091 | 0.000 |
| | Tradition | 3.38 | 1.117 | | | |
| | Complementary activities | 3.18 | 1.189 | | | |

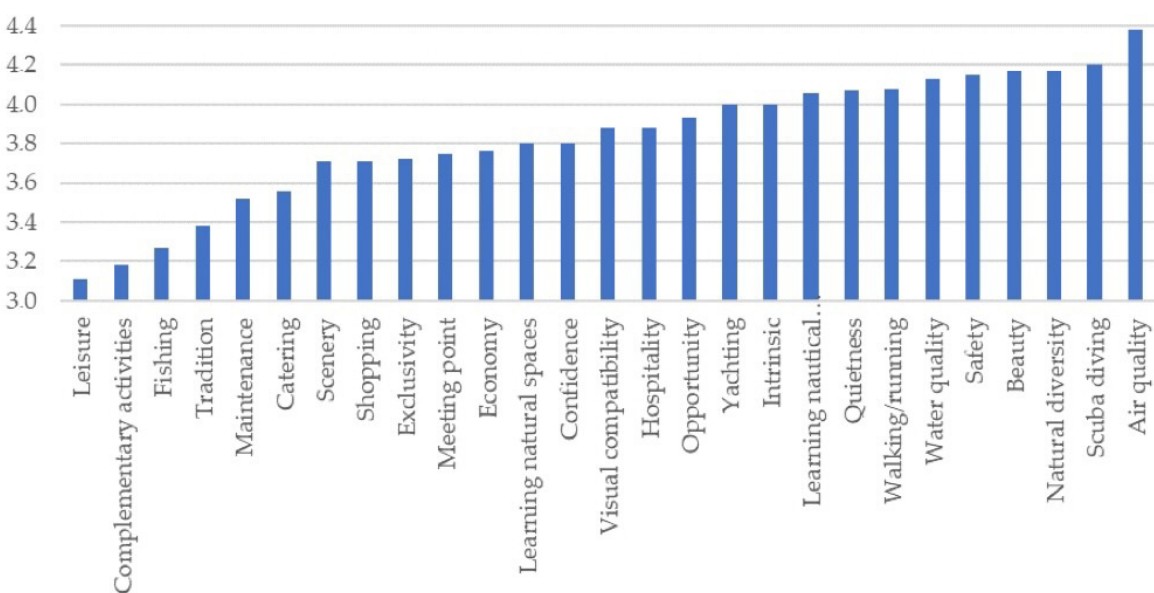

**Figure 3.** Ratings for landscape values.

The basic statistics (mean, standard deviation, standard error or the minimum, maximum, and the upper and lower 95% values of the mean) of each of the 27 landscape values considered for the marina landscape are shown in Table 4 and A3. The values that were agreed more strongly are as follows (Figure 3): issues related to environmental quality—"Air quality" (4.38), "Natural diversity" (4.17), and "Water quality" (4.11) "Scuba diving" (4.20), "Beauty" (4.17), "Safety" (4.15), "Walking/running" (4.08), "Quiet" (4.07), "Learning of nautical activities" (4.06), and "Yachting" (4.00). However, the values which respondents rated at the maximum value (5) at least half of the respondents were scuba diving (61.5%) and air quality (51.0%).

The correlation coefficients between different types of values were obtained (see Appendix B) as a first approximation to the outcomes. Five pairs were strongly correlated, with a coefficient above 0.7. They were "air quality/water quality" (0.728), "catering/meeting point" (0.709), "leisure/shopping" (0.751), "confidence/hospitality" (0.741), and "intrinsic/quiet" (0.747).

We also analyzed the appropriateness of considering the item sets as a single category. The ANOVA model determined that the significant categories were environmental quality (4.22) and aesthetic (3.92). These obtained a high rating, with significant internal consistency. On the contrary, it was found that certain groups needed to have adequate reliability because they had Cronbach's alpha values lower than 0.7: "Learning" (0.689), "Recreation" (0.649), and "Future" (0.471). Nevertheless, groups with high F value and low significance (Sig. < 0.05) means that there were likely differences between the values that comprised it. Therefore, the items should be analyzed individually rather than as group members. Categories with different values, or even reducing some initially considered values, were necessary. These results were consistent with outcomes from PCA. Table 5 shows the results of the component matrix after rotation. Values under 0.5 were excluded (for PCA-based regression model see, Appendix C).

**Table 5.** Component matrix after rotation (PCA).

| Evaluation Item | 1 | 2 | 3 | 4 | 5 | 6 |
|---|---|---|---|---|---|---|
| Leisure | 0.777 | | | | | |
| Catering | 0.771 | | | | | |
| Shopping | 0.765 | | | | | |
| Meeting point | 0.709 | | | | | |
| Walking/running | 0.561 | | | | | |
| Quiet | | 0.844 | | | | |
| Safety | | 0.656 | | | | |
| Intrinsic | | 0.644 | | | | |
| Confidence | | 0.634 | | | | |
| Hospitality | | 0.605 | | | | |
| Exclusivity | | 0.556 | | | | |
| Opportunity | | | | | | |
| Economy | | | | | | |
| Scuba diving | | | 0.746 | | | |
| Yachting | | | 0.737 | | | |
| Learning nautical activities | | | 0.639 | | | |
| Natural diversity | | | 0.630 | | | |
| Fishing | | | 0.527 | | | |
| Visual compatibility | | | | 0.827 | | |
| Scenography | | | | 0.792 | | |
| Beauty | | | | 0.582 | | |
| Tradition | | | | | 0.819 | |
| Maintenance | | | | | 0.694 | |
| Complementary activities | | | | | 0.628 | |
| Learning natural spaces | | | | | | |
| Air quality | | | | | | 0.822 |
| Water quality | | | | | | 0.820 |

Notes: Extraction method; principal component analysis method. Rotation method: Kauser standardized maximum variance method; a rotation was converged after 10 iterations.

Additionally, we compared the landscape values by grouping the respondents based on gender, age, educational level, frequency of visits, seasonal visits, and origin. The ANOVA method was used to test whether there are significant differences concerning these groups. Analysis of variance indicates that there were no significant differences considering gender (1 value). It was also possible to find groups with some differences between the values, such as age (6 values), seasonal visits (7 values), and origin (6 values). Finally, there was a significant difference in educational level (21 values) and frequency of visits (15 values). On the other hand, among the values most likely to be significant differences for the various groups were the economy, fishing, exclusivity, and confidence (4 groups). Likewise, yachting, scuba diving, water quality, natural diversity, learning of natural spaces, and hospitality were pointed out in 3 groups as possible values susceptible to differences.

Concerning the open-ended question, in the first part, we asked what people liked most about the marina. For some respondents, tranquility and the natural environment were central to their arguments. It was encapsulated in responses such as "I like the surroundings, its natural beauty, and tranquility", "I like to walk around there and see the boats [ . . . ], I find it nice", or "What I like most is its location, views, and tranquility". On the contrary, what the respondents liked least about the marina was the excess of urbanization in the nearby areas and the high rates. These arguments have been taken up in replies such as "What I like least is the high prices of the marina, and in particular of fuel", "I do not like the back hill that, by continuing to build, has been destroyed", or "Too much urban development". In addition, we wondered about the images that suggested the marina as a reflection of feelings brought by the perceptions. The most common answers were tranquility, peace, and the Peñón de las Caballas (Figure 4). Other respondents noted sailing, nature, and scuba diving.

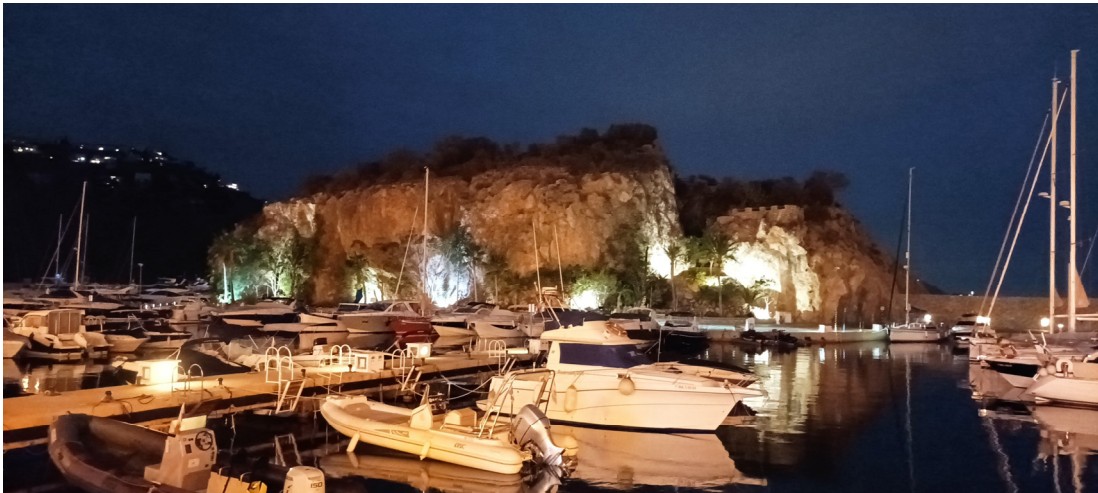

**Figure 4.** View of the Peñón de las Caballas (photograph by authors, October 2022).

Finally, the resume question in this survey was included to assess the importance of landscape within the marina. The response was affirmative in a high percentage.

## 4. Discussion

This study has provided insights into landscape values in marinas. Firstly, it identified the landscape values from interviews and SWOT analysis, which responded to RQ1. Analyzing the results through AMUVAN and PCA methodology represented the basis for responding to RQ2. Finally, the procedure for the study design responded to RQ3 by involving the various parties in determining and rating landscape values considered.

### 4.1. Learnings from SWOT Analysis

SWOT analysis is a qualitative technique that identifies the study area's internal (strengths–weakness) and external (opportunities–threats) factors. As a result of the SWOT analysis of the marina, the marina manager can compare the policies adopted in the marina and the opinion of the stakeholders, with results serving as an indicator of the existing confluence. The stakeholder's opinion, as one of the main assets of the marina, should be taken into account when establishing strategies concerning the landscape and decision-making processes [40]. Moreover, stakeholder engagement brings benefits such as increased public support, facilitating collaborative planning, place-based knowledge, trust, and credibility in the collaborative process [32].

However, several hotspots have been found, and solutions should be solved between all parties. The high fares of services provided by the marina should be justified by excellent service provision and a high level of user satisfaction [4,5,54]. A balance should be sought between improving accessibility to the marina, the capacity of car parks, and a level of overcrowding that does not disturb the central landscape values of the marina, such as tranquility. There was no consensus on whether the marina should be a more exclusive nautical infrastructure or if it should be opened more extensively to the population. Some scholars consider yacht owners to be people with high purchasing power in the search for luxury experiences [54]. Others point out that nautical activities are increasingly attractive, such as nautical chartering, beyond purchasing and maintaining a yacht, which requires a high economic cost [55,56]. In addition, there are no sailing clubs to encourage navigation among the youngest.

Furthermore, the creation of a tradition requires its sustained validity over time [12]. Finally, it must be agreed whether the marina should be a space that only meets nautical needs or also supply the needs not available in nearby nuclei, contributing to improving of the quality of life in the environment [35]. In the latter case, accessibility in its various forms must be guaranteed to enable locals to benefit from such improvements [57].

The interviews reveal that stakeholders base the marina landscape on its appearance. It expresses a need for more awareness of the ELC, requiring more significant effort from competent authorities for its diffusion [58]. However, an implicit understanding of the concept of landscape is held in its broadest sense, even if there is no specific training on the landscape. The fact that the interviews deduce the importance of landscape in Marina del Este results is coherent with the results obtained in previous studies on the economic valuation of the landscape in this same location [59].

Finally, the need for the participation of all parties in landscape management in marinas was evident. All interviewees demanded that their views be taken into account. They disliked that the marina managers did not consider their concerns. However, there was no consensus on how this participation should be channelled. Lack of time, incredibility with a participatory process, and the aversion to holding a meeting without clear results make it necessary for managers to design participatory methods that yield useful outputs, including new technologies for engaging with the public, decreasing costs, and increasing participation and representation [60].

### 4.2. Learnings from Sociodemographic Attributes

Knowing the respondents' profiles was essential to extracting conclusions on how the landscape is perceived. Landscape values are grounded on how people perceive the landscape and the individual and social attributes they assign to it. In this way, most of the respondents were Spanish and residents of the area. Thus, the values assigned were biased by these factors, avoiding a richer assessment if a broader sample (foreigners and visitors from other municipalities) were considered.

The findings in the present study show that the frequency of visits may condition how the landscape is perceived. Respondents who often visit the marina may have a different capacity to admire the environment than those who treat it as an everyday sight. However, those who visit sporadically will feel tremendous admiration.

The results show that most reasons for visiting the marina are sightseeing or sporting activities (51.9%) versus sailing (28.8%). Being a marina, one would expect that the leading cause was sailing. However, this behavior can be explained by twofold causes. First, the surrounding seabed, which begins in the marina, has widespread recognition, so there is a large influx of people attracted by diving. Secondly, marinas are conceived for leisure [12], so services should focus on both nautical users and visitors [14]. Likewise, the marina should become a space of opportunity for the nearby urban centers, providing everything these nuclei lack. In this case, environmental quality and tranquility are two elements that sometimes are impossible to encounter in a coastal holiday area characterized by sun and beach tourism, which increases its population during summer.

On the other hand, the most common way to access the marina is by motor vehicle (53.7%). These results are consistent with the indications of the SWOT analysis, which indicated poor accessibility and parking problems. However, there is a high percentage of users (24.3%) who access the marina on foot. It may be consistent with the fact that sightseeing or some sporting activity is the fundamental reason for visiting. However, it can also be motivated by the fact that many visits were from nearby areas, which makes the marina easily accessible on foot. In any case, only 14.7% of users gain access by sea reflects the need for this group to have little representation in the survey.

There were no significant differences in evaluations by gender. Nonetheless, the values that are most accepted among women are those that are associated with natural aesthetic values, such as "Beauty" (4.08) and "Air quality" (4.21), as well as activities that become well-being, such as "Scuba diving" (4.03), "Walking" (4.00), "Nautical activities" (4.18), "Quiet" (4.10), "Safety" (4.13) or "Opportunity" (4.08). A similar scenario has been illustrated by Ode Sang et al. [61] in the Nordic regions. In this study, women associate higher aesthetic values and well-being outcomes in green spaces with higher naturalness ratings. On the contrary, Kovács et al. [35] point out that many studies indicate that women

enjoy nature less by themselves. However, this behavior does not necessarily apply to another cultural context.

The link between people and places conditions their perception. Attachment to a place is often expressed with pride and a general sense of well-being. The sociodemographic variables that condition attachment are age, social status, education, and residence [23,62]. Additionally, Brown et al. [23] remark in a study from Australia that individuals who live in the coastal zone identify most strongly with smaller areas for their lifestyle and livelihood. The areas of place attachment are smaller. In addition, the older people are, the greater their attachment to their places of residence [38]. This behavior is reflected in the fact that those social groups from the same municipality tend to give higher valuations of an area.

Nevertheless, the group of respondents who visited the marina from a distance of more than 100 km was another with the highest ratings. The effort they make to get there can explain why, therefore, their interest is more remarkable. This interest is reflected in a favorable predisposition.

In another study on peri-urban forests in Japan, there was a high percentage of frequent visits because residents of cities where green spaces cannot meet the residents' needs go to the peri-urban forest to attend to this lack [35]. It cannot explain our study case because the marina is also a working place. It is verified that the minimum evaluations correspond to those who visit the port sporadically, and the largest are usually from the respondents who visit it frequently. It may explain that people less often need to learn how to appropriately value the marina because they have few perceptions of it. On the other hand, those who frequently visit the port may value it more due to a greater attachment.

Concerning the educational level, there is a tendency among respondents (not a generalized rule) that the ratings are lower in those with higher education compared to those who have completed elementary studies. The landscape is formed by the physical elements and the interpretation and transformation the observers make of this reality according to their own experiences [12]. It could be that people with higher education have more refined, specific, or high-level preferences for beauty and nature. Their level of demand is higher when it comes to value. However, with only the data on the education level, it was impossible to draw more conclusions.

Sociodemographic factors have not conditioned the general assessment of the importance of the landscape in the marina. There have been no significant differences in the various groupings. As with the interviewees in the SWOT analysis, the importance of the value of the landscape in the marina is clear.

### 4.3. Learnings from Landscape Values

Globally, humans and landscape perceptions are highly variable. The perception of landscape is a liquid demand insofar as it becomes a service when people request it. In addition, this perception is based on personal experiences, preferences, or abilities [36,63]. This variability in perception or lack of definition makes it difficult to generalize landscape values. In addition, the experience of the place influences values that express affinity with specific values and depend on the local context [26].

Generally speaking, coastal areas are recognized by their scenic and recreational values. However, a geographical location provides contextual nuance. The coastal landform conditions the values being perceived. The proximity of remarkable mountain terrain provides other non-coastal recreation, and, on the opposite, smooth coastal areas are principal sources of recreation and scenic views. In this sense, aesthetic values were the most important ones in coastal landscapes in some countries, such as Australia and Malaysia, but not in other countries, such as Norway, Alaska, or New Zeeland [64]. Floyd et al. [65] point out the natural structure, landscape potential, microclimatic features, and the different activities capable of being developed in the sea as the most crucial factors in the recreational preference towards the city coast. In coastal zones, economic and social values are strongly associated with artificial rather than natural areas [64]. It was also found by Brown and Webber [66], who detected greater importance for economic and recreation values versus

intrinsic, spiritual, and therapeutic values in new coastal development in South Australia. In any case, coastal values are limited by natural dynamics, capacity constraints from pressure from human development, the degradation of ecological functions, and impacts related to climate change [64,67].

Marinas, as infrastructures on the coast, make the most of the existing natural conditions, inserting themselves into the landscape without renouncing their artificial nature but maintaining the prominence of the coast [12]. In this sense, marinas should provide additional values that enhance those which naturally exist in coastal areas. Moreover, a study developed by the United Nations Conference on the impacts of COVID-19 projected growth in preferences for outdoor experiences and contact with nature and water [68]. It supposes a competitive advantage for the marina landscape. Firstly, these facilities are foundations for the perception of the coastal landscape. Secondly, they enhance existing landscape values by adding new ones.

As outcomes from the survey, natural environmental values, beauty, and sports activities were the highest-scoring values (Figure 3 It aligns with the results of Meo et al. [53], who considered the natural environment, scenery, and relaxed lifestyle to be highly valued by residents of coastal areas. It can be argued that the existence of protected natural areas surrounding the port and the importance of La Herradura seabed for scuba diving justify the greater importance of these values. Tranquility is a desirable characteristic of the recreation landscape, as individuals recognize the need for spaces to enhance their general well-being [69]. "Safety" and "Quiet" were two appreciated values perceived by the respondent in the marina.

Nautical tourism strongly impacts both the economy and employment [54,55,70,71]. Moreover, some authors associate coastal developments with greater importance of economic and social values [64,66]. The fact that the "Economy" value scores low reinforces that respondents appreciate the natural character of the landscape. It also may explain low "Visual compatibility" and "Scenery" rates. These values were integrated into the "Aesthetic" group, and ANOVA detected differences between means (high rate for "Beauty" and low rate for the rest). It could be explained by the fact that respondents may have seen a contradiction.

On the one hand, they found a privileged natural environment at the seaside. However, on the land side, they appreciated an excessive urban development that needed more coherence in density and aesthetics (Figure 5). Marinas have served as the justification for real estate projects [12,72]. Excessive urbanization leads to more extensive occupation and degradation of the natural environment along the coastal strip. The neglect of this area leads to a decrease in the valuation of the marina landscape [14].

Within this set of values, there are some in which it is necessary to improve their evaluation because of the importance they have for managing the marina. The principal function of a marina is hospitality [58,59,73,74]. Marinas are infrastructures related to nautical tourism [56,72], and tourism and hospitality are strongly linked [75]. Thus, this value should be among the highest scores. However, the fact that many of the respondents have the purpose of walking means they have little contact with the port staff, so their perception of this value is unrealistic. This explanation is also applied to "Confidence" value. The value "Maintenance" also should be improved, not only for the correct functioning of the elements but also for their significance (coherence between the element and its meaning), as well as the visual impact they can have on visitors and their perceptions of the state of the port [14].

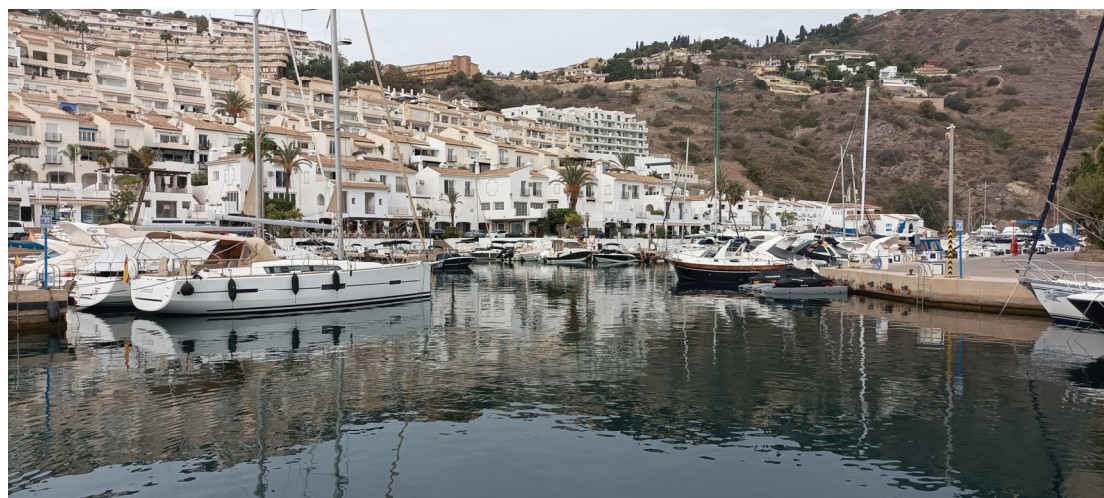

**Figure 5.** View from the port, with urbanization in the background (photograph by authors, October 2022).

Between the lowest values were social ones ("Meeting point", "Catering", "Complementary activities", "Leisure", or "Shopping"), as well as "Tradition" and "Fishing". On the one hand, the results of the interviewees are consistent with the outcomes of the interviews. The lack of commercial development appreciated in the SWOT analysis justifies the low valuation of social aspects. On the other hand, tradition requires a sustained activity over an extended period, avoiding the marina being reduced to mere craft-berthing, adopting monotonous and anodyne solutions, and lacking links to their environment [1]. Finally, fishing is decreasing as restrictions have been increased in nearby protected natural areas.

Tranquility and beauty are, by far, the predominant values and also define the most significant images of the marina retained by visitors. These answers to the open-ended questions were coherent with the values assessments. The concepts of tranquility and the natural environment are the opposite of excessive urbanization, which brings more bustle and a more significant environmental impact. On the contrary, high rates and the lack of maintenance were general claims.

The results reaffirm the need to consider landscape as essential for marina management. However, it would be necessary to determine the level of need for nature/artificiality at which the landscape is considered valuable.

### 4.4. Limitations and Future Research

Interviews and surveys are powerful tools for sound decision-making because they give clear insights into users' preferences. Nevertheless, they are time-consuming to administer and often susceptible to biased interpretation due to insufficient samples that may prevent their generalization [63]. In this sense, a significant limitation of this study was the limited size and the lack of representativeness of the study in terms of foreigners and regular users of the marina. The participants were primarily Spanish and residents of the municipality itself, contrasting with the high number of resident foreigners in the zone. In addition, the most common reason for the visit to the marina is that the walk can tarnish the very function of the marina, which is to serve the boats.

The landscape values assessments should inform the marina manager about the social acceptance of adopted decisions. However, a fundamental limitation of the method is that most values have no direct and or only an unambiguous relationship to the wide range of usage decisions. As Brown et al. [76] put it, to increase the assessment of social landscape, values need to be combined with other spatial measures, such as space use preferences, to provide a fuller explanation of the complex nature of human-landscape valuation and reduce the ambiguity and subjectivity of interpreting place values in decision support.

From the analysis of the survey results using the ANOVA and PCA methods, it is concluded that the values to be considered should be studied in greater detail. It is possible to eliminate some of the values initially considered. Future research should consider a more in-depth analysis of landscape values to be considered. It could strengthen quantitative methods (such as increasing the number of interviews and focus groups) to reach diverse stakeholders and thus improve the targeting of values to ask. The participation of underrepresented groups should also be enhanced.

Personal practices influence landscape values. All information collected from participants is intriguing, even if personal perceptions can influence them [77]. Personal norms are internal personal obligations that condition behavior [78,79]. People's basic mentality (values) and, therefore, their behavior condition how they perceive their environment. In this sense, it is possible to select landscape values that concern the importance of people based on their personal norms.

## 5. Conclusions

This study aimed to apply the landscape value framework to evaluate a marina. We provided a set of landscape values obtained from previous SWOT analysis and performed assessments via a questionnaire conducted with port users. The suitability of the values was analyzed using ANOVA and PCA methods. A survey was carried out, collecting outcomes from 104 users and visitors.

From the analysis of the survey results, the ANOVA method determined that grouping the items into categories was impossible. It was necessary to deal with the values individually. Likewise, the PCA method established that it was possible to make other different groupings and even reduce the number of items.

Considering the data collection, first, the SWOT analysis enabled us to collect the stakeholders' perceptions of management in an orderly and detailed way. The primary outcomes were that the marina had a privileged location but that there was a lack of space, and an excess of urbanization in the surroundings. A marina should be in keeping with an atmosphere of tranquility and well-being. Interviewees also pointed out the need for coordination between agents related to management. This analysis should include more than just the mere collection of information. The comparison with the management of the marina is closely related to the expression of the degree of satisfaction about it, in addition to representing the reflection of the specific needs of this group.

Second, the survey gathered the marina's landscape values. Looking at the overall sample, we found that respondents valued the natural environment, beauty, and sports activities. Nevertheless, there was a need to improve values related to nautical tourism, such as hospitality and maintenance. Other outcomes from the survey were consistent, corroborating the above-related analysis and including high rates and excessive urbanization in the surroundings. Finally, there was a common opinion that the landscape is essential to the marina. Marina managers should consider these outcomes and analyze the points of necessary improvement to establish the causes of these disagreements and propose solutions per the established management model.

Understanding people's motivations and expectations, as well as stakeholder opinions and criteria, are of pivotal importance. The success of achieving a landscape in a marina depends on a delicate interplay of contents and links. Accordingly, the decision-making processes in landscapes require adopting a multi-scale approach. Such an approach needs to encompass the qualities of the landscape that are important to the in-group (manager and stakeholders), distinguishing them from the out-group and thereby uniting people at this level. Their perception can enable more consensual policies to be developed, with greater local acceptance and involvement. A knowledge of weaknesses and threats may align marina managers with user needs more.

This study also demonstrates that public participation is helpful to landscape management in marinas. Public participation is a way to articulate people's preferences, considering them with a genuine interest in managing processes. The study proposes a methodology

to capture this contribution. Moreover, it shows that management is a delicate interplay between the managers' desires, those of the stakeholders, and the way users perceive this landscape. Ultimately, the landscape should be about investing in local values. Landscape in marinas must be distinct from the discourse of the global versus local levels. Increased globalization leads to standards in the quality of services, which creates new scales of abstraction that help people relate to the local features. Furthermore, it has shown the potential to make local landscapes obsolete when activities and products are no longer competitive in the global market.

**Author Contributions:** Conceptualization, R.M. and V.Y.; methodology, R.M. and V.Y.; software, R.M.; validation, R.M. and V.Y.; formal analysis, R.M.; investigation, R.M.; resources, V.Y.; data curation, R.M.; writing—original draft preparation, R.M.; writing—review and editing, R.M. and V.Y.; visualization, R.M.; supervision, V.Y.; project administration, V.Y.; funding acquisition, V.Y. All authors have read and agreed to the published version of the manuscript.

**Funding:** This research was funded by Ministerio de Ciencia e Innovación grant number PID2020-117056RB-I00.

**Institutional Review Board Statement:** Not applicable.

**Informed Consent Statement:** Not applicable.

**Data Availability Statement:** The data presented in this study are available on request from the corresponding author.

**Acknowledgments:** The authors acknowledge the Grant PID2020-117056RB-100 funded by MCIN/AEI/10.13039/501100011033 and by "ERDF A way of making Europe". We also acknowledge all those professionals and anonymous people who agreed to participate in the survey. This study would not have been possible without the contribution of their time, interest, effort, and knowledge.

**Conflicts of Interest:** The authors declare no conflict of interest.

## Appendix A

The interviews featured three parts. The guidelines followed are show in Table A1. We attempted to make the interview more of a guided conversation than a closed list of questions. In this way, participants were allowed to freely express their opinions.

**Table A1.** Marina landscape values' definition.

| Part | Description |
|---|---|
| SWOT | What do you like most about the marina? |
| | What do you like least about the marina? |
| | What would you do to improve marina management? |
| Landscape | How do you define "landscape"? |
| | Do you know the European Landscape Convention (ELC)? |
| | The ELC defines "landscape" as an area, as perceived by people, whose character is the result of the action and interaction of natural and/or human factors. Do you agree with it? |
| Public participation | Would you like to have more participation in the marina management processes? |
| | How would you improve this participation? |

**Table A2.** Marina landscape values' definitions.

| Landscape Value | Description |
|---|---|
| **Aesthetic** | |
| Beauty | I value this place for the perfection of what I see, which is pleasing to the eye. |
| Scenery | I value this place for the composition of the constructions, buildings, and materials, creating a harmonious balance of silhouettes and proportions. |
| Visual compatibility | I value this place because it blends in with its surroundings. |
| **Economy** | I value this place because it provides value and employment in the environment. |
| **Recreation** | |
| Yachting | I value this place for its conditions to sail. |
| Scuba diving | I value this place because of the practice of scuba diving. |
| Fishing | I value this place for fishing. |
| Walking/sunning | I value this place because it allows me to go for walks or run through it. |
| **Environmental quality** | |
| Water quality | I value this place for its clean and crystalline waters. |
| Air quality | I value this place for the fresh and pure air, with the characteristic smell of the sea. |
| Natural diversity | I value this place for the variety of different species that can be seen. |
| **Learning** | |
| Nautical activities | I value this place because it allows the learning of nautical activities. |
| Natural spaces | I value this place because it provides information on nearby natural areas and their conservation. |
| **Social** | |
| Meeting point | I value this place because it provides a meeting point for family and friends. |
| Catering | I value this place because of its gastronomic and bar offer. |
| Shopping | I value this place for its range of stores and services. |
| Leisure | I value this place for the leisure and entertainment opportunities it offers. |
| Exclusivity | I value this place because it is discreet and reserved, with an absence of frivolous, irresponsible, or indecent behavior. |
| **Therapeutic** | |
| Quietness | I value this site because I feel carefree and relaxed. |
| Hospitality | I value this site because the treatment is pleasant and familiar. |
| Confidence | I value this place because the people who work at the marina give me confidence. |
| Safety | I value this place for its sense of safety, with an absence of crime. |
| **Intrinsic** | I value this place for its own sake, for no other reason. |
| **Future** | |
| Opportunity | I value this site because it presents future possibilities for improvement. |
| Maintenance | I value this site because the facilities are kept in a perfect state of conservation and maintenance. |
| **Cultural** | |
| Tradition | I value this place because it has a recognizable maritime past, and it allows a tradition to be passed on. |
| Complementary activities | I value this site because it allows exhibitions, fairs, or similar events. |

# Appendix B

**Table A3.** Descriptive statistics for landscape values.

| | Beauty | Scenery | Visual Compatibility | Economy | Yachting | Scuba Diving | Fishing | Walking/ Running | Water Quality | Air Quality | Natural Diversity | Nautical Activities | Natural Spaces |
|---|---|---|---|---|---|---|---|---|---|---|---|---|---|
| Beauty | 1.000 | | | | | | | | | | | | |
| Scenery | 0.575 | 1.000 | | | | | | | | | | | |
| Visual compatibility | 0.546 | 0.756 | 1.000 | | | | | | | | | | |
| Economy | 0.522 | 0.496 | 0.390 | 1.000 | | | | | | | | | |
| Yachting | 0.475 | 0.211 | 0.339 | 0.385 | 1.000 | | | | | | | | |
| Scuba diving | 0.302 | 0.073 | 0.112 | 0.249 | 0.446 | 1.000 | | | | | | | |
| Fishing | 0.375 | 0.227 | 0.161 | 0.408 | 0.364 | 0.459 | 1.000 | | | | | | |
| Walking/running | 0.479 | 0.283 | 0.271 | 0.396 | 0.097 | 0.224 | 0.302 | 1.000 | | | | | |
| Water quality | 0.432 | 0.266 | 0.299 | 0.313 | 0.311 | 0.390 | 0.425 | 0.354 | 1.000 | | | | |
| Air quality | 0.461 | 0.264 | 0.253 | 0.331 | 0.352 | 0.336 | 0.469 | 0.364 | 0.728 | 1.000 | | | |
| Natural diversity | 0.469 | 0.343 | 0.375 | 0.389 | 0.457 | 0.423 | 0.430 | 0.310 | 0.502 | 0.529 | 1.000 | | |
| Nautical activities | 0.425 | 0.228 | 0.400 | 0.428 | 0.550 | 0.420 | 0.407 | 0.315 | 0.359 | 0.374 | 0.632 | 1.000 | |
| Natural spaces | 0.458 | 0.404 | 0.400 | 0.431 | 0.323 | 0.313 | 0.367 | 0.374 | 0.439 | 0.501 | 0.644 | 0.506 | 1.000 |
| Meeting point | 0.402 | 0.334 | 0.363 | 0.495 | 0.115 | 0.061 | 0.341 | 0.522 | 0.266 | 0.370 | 0.405 | 0.413 | 0.458 |
| Catering | 0.424 | 0.443 | 0.429 | 0.493 | 0.225 | 0.080 | 0.335 | 0.508 | 0.304 | 0.396 | 0.392 | 0.462 | 0.442 |
| Shopping | 0.277 | 0.332 | 0.303 | 0.418 | 0.238 | 0.160 | 0.296 | 0.394 | 0.288 | 0.216 | 0.245 | 0.338 | 0.384 |
| Leisure | 0.308 | 0.421 | 0.342 | 0.387 | 0.101 | 0.069 | 0.179 | 0.441 | 0.301 | 0.265 | 0.301 | 0.252 | 0.464 |
| Exclusivity | 0.359 | 0.329 | 0.381 | 0.407 | 0.430 | 0.405 | 0.402 | 0.419 | 0.363 | 0.321 | 0.340 | 0.507 | 0.270 |
| Quietness | 0.574 | 0.367 | 0.433 | 0.449 | 0.378 | 0.374 | 0.394 | 0.563 | 0.386 | 0.429 | 0.319 | 0.483 | 0.315 |
| Hospitality | 0.510 | 0.389 | 0.360 | 0.551 | 0.452 | 0.340 | 0.327 | 0.375 | 0.272 | 0.345 | 0.377 | 0.519 | 0.444 |
| Confidence | 0.516 | 0.359 | 0.374 | 0.545 | 0.420 | 0.241 | 0.342 | 0.432 | 0.311 | 0.306 | 0.351 | 0.434 | 0.467 |
| Safety | 0.487 | 0254 | 0.283 | 0.365 | 0.370 | 0.333 | 0.340 | 0.421 | 0.426 | 0.504 | 0.366 | 0.302 | 0.263 |
| Intrinsic | 0.687 | 0.406 | 0.452 | 0.585 | 0.461 | 0.368 | 0.473 | 0.570 | 0.453 | 0.558 | 0.437 | 0.551 | 0.484 |
| Opportunity | 0.403 | 0.297 | 0.290 | 0.495 | 0.241 | 0.287 | 0.376 | 0.472 | 0.303 | 0.288 | 0.332 | 0.404 | 0.357 |
| Maintenance | 0.385 | 0.445 | 0.435 | 0.323 | 0.202 | 0.122 | 0.165 | 0.307 | 0.417 | 0.404 | 0.328 | 0.333 | 0.546 |
| Tradition | 0.477 | 0.315 | 0.312 | 0.456 | 0.204 | 0.190 | 0.217 | 0.370 | 0.203 | 0.282 | 0.280 | 0.343 | 0.471 |
| Complementary activities | 0.338 | 0.411 | 0.309 | 0.475 | 0.233 | 0.063 | 0.315 | 0.371 | 0.260 | 0.316 | 0.321 | 0.316 | 0.525 |

**Table A3.** *Cont.*

| | Beauty | Scenery | Visual Compat-ibility | Economy | Yachting | Scuba Diving | Fishing | Walking/Running | Water Quality | Air Quality | Natural Diversity | Nautical Activities | Natural Spaces | |
|---|---|---|---|---|---|---|---|---|---|---|---|---|---|---|
| | Meeting point | Catering | Shopping | Leisure | Exclusivity | Quiet | Hospitality | Confidence | Safety | Intrinsic | Opportunity | Maintenance | Tradition | Complementary usus |
| Meeting point | 1.000 | | | | | | | | | | | | | |
| Catering | 0.709 | 1.000 | | | | | | | | | | | | |
| Shopping | 0.503 | 0.646 | 1.000 | | | | | | | | | | | |
| Leisure | 0.519 | 0.589 | 0.751 | 1.000 | | | | | | | | | | |
| Exclusivity | 0.292 | 0.442 | 0.424 | 0.368 | 1.000 | | | | | | | | | |
| Quietness | 0.379 | 0.431 | 0.313 | 0.294 | 0.693 | 1.000 | | | | | | | | |
| Hospitality | 0.503 | 0.474 | 0.470 | 0.430 | 0.569 | 0.689 | 1.000 | | | | | | | |
| Confidence | 0.417 | 0.471 | 0.383 | 0.435 | 0.523 | 0.637 | 0.741 | 1.000 | | | | | | |
| Safety | 0.380 | 0.433 | 0.366 | 0.319 | 0.441 | 0.629 | 0.544 | 0.543 | 1.000 | | | | | |
| Intrinsic | 0.566 | 0.628 | 0.394 | 0.350 | 0.528 | 0.747 | 0.642 | 0.614 | 0.602 | 1.000 | | | | |
| Opportunity | 0.392 | 0.367 | 0.394 | 0.458 | 0.552 | 0.567 | 0.425 | 0.444 | 0.333 | 0.568 | 1.000 | | | |
| Maintenance | 0.326 | 0.358 | 0.420 | 0.484 | 0.388 | 0.296 | 0.386 | 0.314 | 0.275 | 0.422 | 0.335 | 1.000 | | |
| Tradition | 0.357 | 0.350 | 0.394 | 0.432 | 0.303 | 0.389 | 0.512 | 0.521 | 0.287 | 0.436 | 0.445 | 0.591 | 1.000 | |
| Complementary activities | 0.503 | 0.540 | 0.547 | 0.512 | 0.345 | 0.249 | 0.468 | 0.470 | 0.265 | 0.446 | 0.405 | 0.578 | 0.600 | 1.000 |

## Appendix C

Once the final factorial solution has been reached, it is often interesting to obtain an estimate of the subjects' scores on each of the resulting factors. The coefficient matrix of factor scores (Table A4) allows us to obtain a table with the weights necessary to calculate the factor scores from the original variable.

**Table A4.** Component score coefficient matrix.

| Evaluation Item | 1 | 2 | 3 | 4 | 5 | 6 |
|---|---|---|---|---|---|---|
| Beauty | −0.174 | 0.130 | −0.082 | 0.231 | 0.081 | 0.025 |
| Scenery | 0.024 | −0.085 | −0.036 | 0.421 | −0.119 | −0.031 |
| Visual compatibility | −0.035 | −0.058 | −0.004 | 0.447 | −0.126 | −0.040 |
| Economy | 0.039 | 0.031 | 0.103 | 0.022 | 0.039 | −0.122 |
| Yachting | −0.147 | −0.020 | 0.328 | 0.058 | 0.027 | −0.113 |
| Scuba diving | 0.009 | −0.054 | 0.328 | −0.128 | −0.067 | 0.000 |
| Fishing | 0.142 | −0.014 | 0.174 | −0.164 | −0.141 | 0.080 |
| Walking/running | 0.166 | 0.126 | −0.154 | −0.115 | −0.041 | 0.089 |
| Water quality | −0.062 | −0.041 | −0.044 | −0.032 | 0.005 | 0.449 |
| Air quality | −0.053 | 0.013 | −0.091 | −0.039 | −0.015 | 0.454 |
| Natural diversity | −0.004 | −0.204 | 0.242 | 0.076 | 0.001 | 0.132 |
| Learning nautical activities | −0.042 | −0.063 | 0.256 | 0.103 | 0.013 | −0.097 |
| Learning natural spaces | 0.009 | −0.207 | 0.107 | 0.074 | 0.184 | 0.104 |
| Meeting point | 0.261 | −0.055 | −0.035 | 0.059 | −0.135 | −0.007 |
| Catering | 0.300 | −0.051 | −0.046 | 0.091 | −0.210 | −0.002 |
| Shopping | 0.305 | −0.076 | 0.041 | −0.070 | −0.074 | −0.078 |
| Leisure | 0.284 | 0.092 | −0.047 | −0.052 | 0.004 | −0.010 |
| Exclusivity | 0.049 | 0.143 | 0.136 | −0.039 | −0.118 | −0.126 |
| Quietness | −0.101 | 0.342 | −0.081 | −0.019 | −0.072 | 0.021 |
| Hospitality | −0.043 | 0.165 | 0.075 | −0.016 | 0.079 | −0.159 |
| Confidence | −0.060 | 0.193 | 0.023 | −0.026 | 0.091 | −0.016 |
| Safety | −0.028 | 0.287 | −0.182 | −0.084 | −0.117 | 0.218 |
| Intrinsic | −0.060 | 0.192 | −0.087 | 0.086 | −0.012 | 0.049 |
| Opportunity | 0.091 | 0.129 | 0.036 | −0.175 | 0.046 | −0.084 |
| Maintenance | −0.104 | −0.062 | −0.110 | 0.021 | 0.389 | 0.123 |
| Tradition | −0.134 | 0.052 | −0.037 | −0.158 | 0.521 | −0.069 |
| Complementary activities | 0.089 | −0.075 | −0.006 | −0.110 | 0.301 | −0.051 |

Notes: Extraction method; principal component analysis method. Rotation method: Varimax with Kauser normalization method; rating of components.

In addition to the factor scores, the variance-covariance matrix of the factor scores (Table A5) indicates that the factor scores of the various factors are completely independent of each other.

**Table A5.** Variance-covariance matrix of factor scores.

| Components | 1 | 2 | 3 | 4 | 5 | 6 |
|---|---|---|---|---|---|---|
| 1 | 1.000 | 0.000 | 0.000 | 0.000 | 0.000 | 0.000 |
| 2 | 0.000 | 1.000 | 0.000 | 0.000 | 0.000 | 0.000 |
| 3 | 0.000 | 0.000 | 1.000 | 0.000 | 0.000 | 0.000 |
| 4 | 0.000 | 0.000 | 0.000 | 1.000 | 0.000 | 0.000 |
| 6 | 0.000 | 0.000 | 0.000 | 0.000 | 1.000 | 0.000 |
| 6 | 0.000 | 0.000 | 0.000 | 0.000 | 0.000 | 1.000 |

Notes: Extraction method: Principal components. Component scores.

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
