# Peer review of "Landscape Values in a Marina in Granada (Spain): Enhancing Landscape Management through Public Participation"

_land, doi:10.3390/land12020492_

Round 1

Reviewer 1 Report (Previous Reviewer 2)

I commented on this manuscript several days ago. 

After R1, the editors withdrew the manuscript and resubmitted again.

The quality of this manuscript got better.

But there were still some issues.

The key issues with this manuscript were still needed to focus on my prev"ous comments:

The value and purpose of this manuscript is not "public participation". Finding a way or a process to improve the marina’s landscape values through "public participation" which is the true value of this study.

The paper presents some interesting contextual data but simply reconfirms existing knowledge and relations in a new context. The authors need to revise from a target not with the "public participation".

Author Response

Point A0.    After R1, the editors withdrew the manuscript and resubmitted again. The quality of this manuscript got better.

We are pleased that the reviewer appreciated the effort made by the authors in following his/her observations. We hope to adapt these new corrections to the guidelines indicated.

Point A1.    The value and purpose of this manuscript is not "public participation". Finding a way or a process to improve the marina’s landscape values through "public participation" which is the true value of this study.

The authors would like to thank the reviewer for his dedication and time. The purpose of the study, as reflected in the research questions (RQ), is to determine the values of the landscape. It proposes a methodology to collect data and procedures to verify the accuracy of the outcomes. Collaterally, the procedure followed to perform the valuation also represents a form of public participation in the landscape. This is therefore a secondary issue, but the results may be of interest for the management of the marina. In this way, the title and the introduction have been rewritten.

The title now reads: “Landscape values in a marina in Granada (Spain): Enhancing landscape management through public participation”.

The text now reads (section 1): The management is responsible for controlling an organization and dealing with it carefully. This study focuses how to identify and assess landscape values in a marina through public participation. It also represents an opportunity to enhance the involvement of stakeholder and public in marina’s landscape management solutions”.

The text now reads (section 1.3): This study seeks to apply the landscape value framework to the evaluation of a marina. The limited landscape research in marinas is why the landscape values in these maritime facilities have yet to be studied. The study was carried out though public participation, which provides outcomes from a theoretical framework to improve marinas' landscape management”.

Point A2.    The paper presents some interesting contextual data, but it simply reconfirms existing knowledge and relations in a new context. The authors need to revise from a target that is not with the "public participation".

We appreciate all reviewer comments that can be used to improve the study. As discussed in section 1.1, landscape values have been applied in several areas, but not in marinas. The novelty of this study represents the study of landscape values in an area where no previous work has been done.

As discussed in the previous point, public participation is a collateral element. It represents a tool through which the study has been carried out.

The authors reiterate their gratitude for the comments made by the reviewer, as they have improved the clarity and understanding of the manuscript.

Reviewer 2 Report (New Reviewer)

Landscape values are a very important aspect shaping the space. Their perception by users has a large impact on their mental and physical well-being. Therefore, I believe that the subject matter taken up by the authors is important and worth publishing, but I would like to point out a few elements that would require additional consideration by the authors:

L27 - 109

The introduction presents the research background referring to the research object and methodology. It would be worth supplementing this chapter with a reference to European and global legal regulations related to shaping the landscape and space. In addition, it would be worthwhile to develop legal issues related to the shaping and designing of marinas, especially since an important element in the implementation of this type of facilities is water management, which may also affect the form of the tested facility.

L390 - 616

The article contains a very extensive discussion, which blurs the research results and should be presented in a shorter, but more synthetic and concrete way.

L617-664

Conclusions would also need to be shortened and made more specific. A great advantage is the presentation of recommendations, but they should be described in a more synthetic way.

Author Response

Point B0.    Landscape values are a very important aspect shaping the space. Their perception by users has a large impact on their mental and physical well-being. Therefore, I believe that the subject matter taken up by the authors is important and worth publishing.

We thank the reviewer for his/her interest and support for the chosen topic of the study.

Point B1.    The introduction presents the research background referring to the research object and methodology. It would be worth supplementing this chapter with a reference to European and global legal regulations related to shaping the landscape and space. In addition, it would be worthwhile to develop legal issues related to the shaping and designing of marinas, especially since an important element in the implementation of this type of facilities is water management, which may also affect the form of the tested facility.

We would like to acknowledge the reviewer's comment, which improves the study. As the reviewer points out, a global reference to landscape could be included. Although landscape is an element on which various legal texts have had an impact, either directly or indirectly, at the international level, it was not until 2000 that a Convention dealing specifically with this issue was concluded. The European Landscape Convention, sponsored by the Council of Europe and approved in Florence in October 2000, is the only international treaty focused exclusively on landscape; until then, landscape had been dealt with intuitively, recognising the need to preserve the environment as a guarantor of the maintenance of people's quality of life, but in a biased and sectoral manner, focusing on its natural and cultural aspects.

In 1872, the first national park was created in the United States, which represented the materialisation of conservationism, concretised in the conservation of the beauty of an extraordinary place based on primarily aesthetic criteria, its grandeur, beauty or singularity, requiring the protection of these exceptional and unspoilt areas, and whose social purpose was the preservation of aesthetic contemplation and reflection. This meant the acceptance of naturalist ideas and the introduction of romanticism, allowing a renewal of the vision of nature, in which the beauty of landscapes is to be found.

In 1913, the First International Conference on the Protection of Natural Landscapes was held in Bern, which was the first attempt to address this issue by laying the foundations of protectionism, reflecting the relationship between landscape and nature, but it could not be developed due to the outbreak of the First World War, which meant that in the end it was of no importance.

The Athens Charter - City Planning Charter - is an urban planning manifesto devised at the 4th International Congress of Modern Architecture in 1933 and released in 1942 by Le Corbusier. This document hints at the concept of landscape without actually mentioning it by calling for respect for the character and physiognomy of cities and the need for pleasant visual elements, as well as referring to the environment with terms such as "surroundings" or "proximities".

The 1972 United Nations Conference on the Environment in Stockholm was the most important initiative for landscape and environmental conservation, recognising the mutual influence between man and his environment; "man is both the work and the maker of the environment around him, which provides him with material sustenance and the opportunity for intellectual, moral, social and spiritual development". Within the 26 points approved in this Declaration, reference is made to the enjoyment of a quality environment as a fundamental right of people, at the same time as sustainable development -both economic and social- and the preservation of natural ecosystems, for which it is necessary to allocate resources for the conservation and improvement of the environment, taking into account the needs of each country, through an integrated and coordinated approach to planning.

It is within the European framework that landscape is expressly referred to, albeit with an ambiguous and sectoral character. In the European Coastal Charter, formulated and approved in October 1981 at the Conference of Peripheral Maritime Regions of the EEC held in October 1981 in Crete and subsequently ratified by the European Parliament in June 1982, specific reference is made to the coastal landscape. Likewise, within the Council of Europe, the European Spatial Planning Charter, approved by the European Conference of Ministers of Spatial Planning in May 1983 in Torremolinos, mentions the consideration of landscape conservation and management in rural regions, while for urban regions, this concept is mentioned in an ambiguous way, this concept is mentioned in an ambiguous way by considering "architectural heritage, monuments and picturesque sites" as elements of revaluation to be considered in the general policy of spatial and urban planning, which implies a differentiation between natural and cultural landscapes.

Prior to the European Landscape Convention, and on a sectoral basis within the Mediterranean countries as a whole, the Mediterranean Landscape Charter had been drawn up in June 1992, signed by the regions of Languedoc-Roussillon, Veneto (later Tuscany) and Andalusia, taking advantage of the meeting provided by the Universal Exhibition in Seville, It was subsequently adopted in the framework of the Council of Europe, within the Conference of Mediterranean Regions, at its third meeting in Taormina (Italy) in April 1993, and unanimously endorsed by Resolution at the 21st General Assembly of the Conference of Peripheral Maritime Regions of the EEC in Saint-Malo in October 1993. This Charter represented the policy they intended to follow in terms of landscape protection, taking into account the ecological problems and promoting a greater understanding of the risks to which the territory is subject. It defines landscape "as the formal manifestation of the sensitive relationship of individuals and societies in space and time with a territory more or less intensely shaped by social, economic and cultural factors. Landscape is thus the result of the combination of natural, cultural, historical, functional and visual aspects". Landscape is thus recognised as an essential value in the framework of the life and culture of the peoples of Europe, as a resource for individuals, which means that measures must be taken to protect the landscape in order to deal with the aggressions that modern society inflicts on it. This Declaration stated the intention to "implement a land-use planning policy which cooperates in the protection of natural resources, the conservation of historical and cultural assets, the active maintenance of the landscape and the environmental balance". The implementation of landscape policies implies that "this way of understanding the territory and of acting, valuing and respecting its landscape aspect, should be reflected in planning instruments, assessment procedures, technical instructions and projects, in order to achieve a progressive physical integration of all actions with a territorial impact". Therefore, this Declaration is one of the bases for the preparation of the CEP; starting from a concept of landscape and a diagnosis, it incorporates the ideas relating to the preferential connection with the policies with the greatest impact on the landscape, the extension of the objectives on the landscape to its management, going beyond a simple protectionist concept, or the need to identify one's own landscapes, raising awareness among the population, educating and training specialists.

We believe that all this information, although interesting, may distract from the central objective of the study, which is the values of the landscape.

In relation to marinas, there are no aesthetic constraints in the regulatory framework. This issue is addressed in the environmental impact studies, which establish the need to assess landscape impacts as well.

Point B2.    The article contains a very extensive discussion, which blurs the research results and should be presented in a shorter, but more synthetic and concrete way.

We appreciate all reviewer comments that can be used to improve the study. Discussion reviews the findings and put them into the context of the overall research. This section contains interpretation, analysis, and explanations of the results. In addition to synthesising the data, this section has been divided into three subsections to make it easier to read, as well as to group results.

The authors are aware of the length of the discussion. The methodology employed (surveys and questionnaire), the interpretation of the socio-demographic characterisation of the participants, as well as the total number of values considered (27) justify this length. It is also necessary to find a balance with the comments of other reviewers who demand more detail in the explanations.

Point B3.    Conclusions would also need to be shortened and made more specific. A great advantage is the presentation of recommendations, but they should be described in a more synthetic way.

As mentioned in the previous section, the aim is to combine the synthesis of extensive information with the demands made by other reviewers for more information. However, we have attempted to summarize the text.

Conclusions now reads (section 5): This study aimed to apply the landscape value framework to evaluate a marina. We provided a set of landscape values obtained from previous SWOT analysis and assessed through a questionnaire conducted with marina users. A survey was carried out, collecting outcomes from 104 users and visitors.

“From the analysis of the survey results, the ANOVA method determined that grouping the items into categories was not convenient, dealing with the values individually. Likewise, the PCA method established that it was possible to make other different groupings and even reduce the number of items.

“Considering the data collection, first, the SWOT analysis enabled us to collect in an orderly and detailed way the perception of management by the stakeholders. The primary outcomes were that the marina had a privileged location, but there was a lack of space and an excess of urbanization in the surroundings. Marina should be in keeping with an atmosphere of tranquillity and well-being. Interviewees also pointed out the need for coordination between agents related to management.

“Second, the survey gathered the marina’s landscape values. Respondents valued the natural environment, beauty, and sports activities. Nevertheless, there was a need to improve values related to nautical tourism, such as hospitality and maintenance. Other outcomes from the survey were consistent, corroborating the above related and including high rates and excessive urbanization in the surroundings. Finally, there was a common opinion that the landscape is essential to the marina. The landscape should be about investing in local values. Landscape in marinas must be distinct from the global versus local discourse. Increased globalization leads to standards in the quality of services, which creates new scales of abstraction that help people relate to the local features.

“This study provides data to marinas managers to analize their policies. The outcomes are closely related to the expression of the degree of satisfaction of stakeholders and public. Marina managers should consider these results and analyze the points of improvement to establish the causes of these disagreements and propose solutions to the established management model.

“This study also demonstrates that public participation is helpful to landscape management in marinas. It proposes a methodology to capture this contribution, which involves stakeholders and public. Moreover, it shows that management is a delicate interplay between the managers’ desires and the stakeholders, and how users perceive this landscape. Understanding people's motivations and expectations, as well as stakeholder opinions and criteria, are of pivotal importance. The success of achieving a landscape in a marina depends on a delicate interplay of contents and links. Considering public participation may enable more consensual policies with greater acceptance and involvement”.

The authors reiterate their gratitude for the comments made by the reviewer, as they have improved the clarity and understanding of the manuscript.

Reviewer 3 Report (New Reviewer)

The study presented is interesting and detailed.

I suggest improving the quality of the document:

1. incorporate the source of the fig no. 2, 4 y 5

2. Correct the error in reference no. 37. The name of one of the authors is Perez Montero and not Perez Monteiro as it appears.

3. It could help to the comprehension of the text to include a methodological graph.

4. It would be interesting to know data on the number of visitors (e.g. annual average) to the port, which has served as the universe for the sample taken for the study.

5. It would be important to consider explicitly in the conclusions, according to the research findings, the answer to question RQ3. How could the landscape values framework improve public participation in landscape management?

Author Response

Point C0.    The study presented is interesting and detailed.

The authors thank the reviewer for his time, knowledge and efforts to improve the study.

Point C1.     incorporate the source of the fig no. 2, 4 y 5.

We would like to acknowledge the reviewer's comment, which improves the study. Initially, the source of the images was not indicated when they were taken by the authors. Following the reviewer's suggestion, we have added the following text “(photograph by authors, October 2022)”.

Point C2.    Correct the error in reference no. 37. The name of one of the authors is Perez Montero and not Perez Monteiro as it appears.

We appreciate the reviewer's comments. We have corrected the error.

Point C3.    It could help to the comprehension of the text to include a methodological graph.

We would like to thank the reviewer for the detail appreciated. The authors have tried to explain the research methodology in the text. We believe that the inclusion of more images may increase the length of the study, diverting attention from the values of the landscape.

Point C4.    It would be interesting to know data on the number of visitors (e.g. annual average) to the port, which has served as the universe for the sample taken for the study.

The authors are grateful for the reviewer's interest in improving the study. There is no data on the number of visitors to the marina. The minimum number of participants has been determined through statistical criteria. As indicated in the text (lines 301-303): “The precision level depends on the risk a researcher is willing to take. Using a level of confidence of 95% and considering an unlimited population, the margin of error in the proportion is set at 0.1, which is acceptable in perception survey”.

Point C5.    It would be important to consider explicitly in the conclusions, according to the research findings, the answer to question RQ3. How could the landscape values framework improve public participation in landscape management.

We appreciate all reviewer comments. We have rewritten part of the conclusions to reinforce RQ3.

Conclusions now reads (section 5): This study aimed to apply the landscape value framework to evaluate a marina. We provided a set of landscape values obtained from previous SWOT analysis and assessed through a questionnaire conducted with marina users. A survey was carried out, collecting outcomes from 104 users and visitors.

“From the analysis of the survey results, the ANOVA method determined that grouping the items into categories was not convenient, dealing with the values individually. Likewise, the PCA method established that it was possible to make other different groupings and even reduce the number of items.

“Considering the data collection, first, the SWOT analysis enabled us to collect in an orderly and detailed way the perception of management by the stakeholders. The primary outcomes were that the marina had a privileged location, but there was a lack of space and an excess of urbanization in the surroundings. Marina should be in keeping with an atmosphere of tranquillity and well-being. Interviewees also pointed out the need for coordination between agents related to management.

“Second, the survey gathered the marina’s landscape values. Respondents valued the natural environment, beauty, and sports activities. Nevertheless, there was a need to improve values related to nautical tourism, such as hospitality and maintenance. Other outcomes from the survey were consistent, corroborating the above related and including high rates and excessive urbanization in the surroundings. Finally, there was a common opinion that the landscape is essential to the marina. The landscape should be about investing in local values. Landscape in marinas must be distinct from the global versus local discourse. Increased globalization leads to standards in the quality of services, which creates new scales of abstraction that help people relate to the local features.

“This study provides data to marinas managers to analize their policies. The outcomes are closely related to the expression of the degree of satisfaction of stakeholders and public. Marina managers should consider these results and analyze the points of improvement to establish the causes of these disagreements and propose solutions to the established management model.

“This study also demonstrates that public participation is helpful to landscape management in marinas. It proposes a methodology to capture this contribution, which involves stakeholders and public. Moreover, it shows that management is a delicate interplay between the managers’ desires and the stakeholders, and how users perceive this landscape. Understanding people's motivations and expectations, as well as stakeholder opinions and criteria, are of pivotal importance. The success of achieving a landscape in a marina depends on a delicate interplay of contents and links. Considering public participation may enable more consensual policies with greater acceptance and involvement”.

The authors reiterate their gratitude for the comments made by the reviewer, as they have improved the clarity and understanding of the manuscript.

Reviewer 4 Report (New Reviewer)

It is not clear if the purpose is to suggest and manage the global services / activities offer orjust to analyse the management quality of the Marina's managers which referes more directly with human and operational relationhips and not only the services offer.

It is not understandable the interviewees' profile of the qualitaty study. It is only explained the businees connection with the Marina, but not the position of the interviewee, for example are they all the managers? or employees of these firms?

Also not understand the SWOT as a methodological tool. But even if it is aplyed, it should be corrected. The presented SWOT is not well done, because it is written weaknesses twice and not Threats. But namely because the Opportunities indentified are indeed the strategies suggested and not really the opportunities factor. It is very difficult to understand the external factors of that marina landscape. So I suggest to remove the SWOT tool from the article and just explain better the qualitative sample characteristics, their internal analysis (strenghs/weaknesses) and suggestions to improve the marina services offer.

Author Response

Point D1.    It is not clear if the purpose is to suggest and manage the global services / activities offer or just to analyse the management quality of the Marina's managers which refers more directly with human and operational relationhips and not only the services offer.

The authors would like to thank the reviewer for his dedication and time. The purpose of the study, as reflected in the research questions (RQ), is to determine the values of the landscape. It proposes a methodology to collect data and procedures to verify the accuracy of the outcomes. Collaterally, the procedure followed to perform the valuation also provides the marina managers with indirect information on the perception of the quality of the services and the human relations provided.

Point D2.    It is not understandable the interviewees' profile of the qualitaty study. It is only explained the business connection with the Marina, but not the position of the interviewee, for example are they all the managers? or employees of these firms?

We appreciate all reviewer comments. I can be used to improve the qualitaly accuracy of the data. All the persons interviewed were the managers of their businesses. This has been indicated in the text.

The text now reads (section 2.2.1): “Table 1 shows the participants and their affiliations. In order to improve the quality of the responses, the position of all respondents was in charge of their business.

Point D3.    Also not understand the SWOT as a methodological tool. But even if it is applied, it should be corrected. The presented SWOT is not well done, because it is written weaknesses twice and not Threats. But namely because the Opportunities identified are indeed the strategies suggested and not really the opportunities factor. It is very difficult to understand the external factors of that marina landscape. So I suggest to remove the SWOT tool from the article and just explain better the qualitative sample characteristics, their internal analysis (strengths/weaknesses) and suggestions to improve the marina services offer.

We would like to thank the reviewer for the detail appreciated. We have corrected the error detected by the reviewer. The SWOT analysis is not in itself a methodological tool, but one to carry out the methodology proposed. It pulls information internal and external sources that may have impacts to decisions. And management items may be valued from a landscape viewpoint (Martín and Yepes, 2021). Moreover, it also helps to collect the people’s perceptions towards the marina.

Attending to reviewer`s observation, the results of the SWOT analysis have been modified. The opinions collected through the interviews were no correctly ordered. There were elements that corresponded to external factors placed as internal sources and vice versa. Following the reviewer's suggestion, the data have been revised and put in the correct order.

The text now reads (section 2.2.1): The first step was to identify a set of complementary information from interviews to obtain a SWOT analysis. This approach related a twofold objective: firstly, it could be compared with that performed by marina managers, and also related management items to landscape {14}; secondly, it served as a basis for establishing the features of the marina and thus determining the nature of the values on which to question users”.

Table 2 now reads:

Strengths

Weakness

Professional and friendly staff, who convey trust and confidence.

Highest rates in the whole area.

Diving centre of reference in the area.

Poor land transport connections and no nearby urban centers.

Holding championships, tournaments, and fairs.

A small port with limited berths and no possibility of long promenades.

Environmental management procedures.

The existence of disputed areas prevents their exploitation.

Possibility of creating a pedestrian promenade with a nearby city center

Lack of parking space, with congestion problems in the summertime.

Possibility of making development space available when outstanding disputes are cleared up.

Lack of parking space, with congestion problems in the summertime.

Excessive urbanization is growing faster than port services.

Poor land transport links and no nearby urban centres.

Lack of coordination between the marina and the other administrations.

Opportunities

Threats

High berth demand.

Environmental protection constraints for the development of nautical activities.

Idyllic setting, with several protected natural areas nearby.

High seasonality of the tourism product, based on the binomial sun-beach.

Potential for development of nautical and marine-related activities.

Little nautical tradition, with no strategy to promote it among the population.

Good climate with mild winter temperatures and low rainfall.

High exposure to easterly storms.

The existence of unregulated anchorages in the vicinity.

[14]      Martín, R.; Yepes, V. Bridging the gap between landscape and management within marinas: A review. Land 2021, 10, 821. https://doi.org/10.3390/land10080821.

The authors reiterate their gratitude for the comments made by the reviewer, as they have improved the clarity and understanding of the manuscript.

Reviewer 5 Report (New Reviewer)

This paper represents an interesting and well-carried-out research study that attempts to identify and assess landscape values and perceptions in a marina in Granada. The structure, organization, presentation and overall delivery of the research and its outcomes are excellent. Where I had a serious issue was with the language and expression (often unclear and imprecise), making it difficult to understand certain points/ statements in a definitive manner. Please rectify.

The main points I would like to raise have to do with a need for sharper delivery of your points and arguments; they require small interventions and can be easily remedied. First, please select what you intend to be the main objective of your study, between a) identifying and assessing a marina's landscape values (abstract; l.29 etc.) or b) application of the landscape framework to the evaluation of a marina (l.88-89). I assume the main goal is the former, which is broader in scope, as your research questions attest to.

With regard to the research questions (top of p. 3), please take care to show that these questions are to be answered through a users' survey and stakeholders' interviews, because the study simply cannot by itself "enhance public participation in marina's landscape management solutions" (l.30): it may only raise awareness among interested/ involved parties about it.

Furthermore, you may consider adding 'perceptions' (perhaps also 'preferences') to 'values' in RQ1, or emphasize these aspects of the user-landscape relationship, as well (elsewhere?), as you clearly include questions of this sort in your data collection.

You are tackling a very complex type of landscape, which could be characterized as natural, as tourist, as marina, etc. and this makes it a bit more difficult to justify and interpret your findings--which you do very well, nonetheless. However, you should take more into consideration in your discussion that this study involves mostly Spanish visitors and area residents and that their landscape values are not just influenced by personal behaviors (you probably mean personal practices? l. 611), but by a series of factors (perceptions, preferences, norms, values, vested interests etc.), some of which you mention further down, but which are key to the interpretation of the findings--a complex and challenging task, anyway. For instance, concerning the survey respondents' educational level (ls. 494-498), it could be that those with a higher education have more refined, specific or high-standard preferences for beauty and nature (scenic and recreational values) in a landscape than those with a lower education, as marinas are highly artificial environments. You acknowledge the artificiality of the marina later on (top of p.15), but the fact that people may increasingly prefer outdoor/ natural environments does not imply that they will also opt for marinas (ls. 527-533: this whole paragraph seems a bit too speculative). It is indeed a complex question you are coming up against, namely "how artificial do the users of the marina consider its landscape"? The answer, of course, which may help you in the interpretation of your results, depends on a great variety of factors: cultural (i.e. a mostly 'urban culture' or not?); geographical proximity of alternative nature destinations; traditions; dependence of area's livelihoods on the marina, etc., besides, of course, the marina's scenic and recreational values. Just food for thought.

Finally, some smaller points:

1. Please, present Appendix A in the text (ls. 180-190) in a clearer way, step by step, in line with your 2-step data collection scheme (ls.164-165)

2. Specify what you mean by 'high rates' (of what?) both in Table 2 and in the text (l. 583)

3. Insert 'Threats' in Table 2, in place of 'Weaknesses' which appears twice

4. You may consider ending the Conclusions not only with the positive repercussions of 'increased globalization' but also with a reference to its more negative ones, in light of the broader current quest for more sustainable futures

Author Response

Point E0.    This paper represents an interesting and well-carried-out research study that attempts to identify and assess landscape values and perceptions in a marina in Granada. The structure, organization, presentation and overall delivery of the research and its outcomes are excellent.

We thank the reviewer for his/her interest and support for the chosen topic of the study.

Point E1.    Where I had a serious issue was with the language and expression (often unclear and imprecise), making it difficult to understand certain points/ statements in a definitive manner. Please rectify.

The authors would like to thank the reviewer for his interest. The paper has been professionally proofread

Point E2.    The main points I would like to raise have to do with a need for sharper delivery of your points and arguments; they require small interventions and can be easily remedied. First, please select what you intend to be the main objective of your study, between a) identifying and assessing a marina's landscape values (abstract; l.29 etc.) or b) application of the landscape framework to the evaluation of a marina (l.88-89). I assume the main goal is the former, which is broader in scope, as your research questions attest to.

We would like to thank the reviewer for his appreciation. It is true that the main object of the study should be clearer. We must emphasize the identification and valuation of landscape values. In this sense, the first paragraph of section 1.3 has been rewritten, as well as RQ3.

The text now reads (section 1.3): This study seeks to apply the landscape value framework to the evaluation of a marina. The limited landscape research in marinas is why the landscape values in these maritime facilities have yet to be studied. The study was carried out though public participation, which provides outcomes from a theoretical framework to improve marinas' landscape management.

RQ3 now reads (section 1.3): How could public participation in determining landscape values be increased?”.

Point E3.    With regard to the research questions (top of p. 3), please take care to show that these questions are to be answered through a users' survey and stakeholders' interviews, because the study simply cannot by itself "enhance public participation in marina's landscape management solutions" (l.30): it may only raise awareness among interested/ involved parties about it.

The authors would like to thank the reviewer for his dedication and time. In coherence with what was stated in the previous point (E2), we must evidence the purpose of the study. It is determining the values of the landscape in a marina. It proposes a methodology to collect data and procedures to verify the accuracy of the outcomes. Collaterally, the procedure followed to perform the valuation also represents a form of public participation in the landscape. This last aspect has been emphasized by modifying the text of the title and the first paragraph of the introduction (section 1).

The title now reads: “Landscape values in a marina in Granada (Spain): Enhancing landscape management through public participation”.

The text now reads (section 1): The management is responsible for controlling an organization and dealing with it carefully. This study focuses how to identify and assess landscape values in a marina through public participation. It also represents an opportunity to enhance the involvement of stakeholder and public in marina’s landscape management solutions”..

Point E4.    Furthermore, you may consider adding 'perceptions' (perhaps also 'preferences') to 'values' in RQ1, or emphasize these aspects of the user-landscape relationship, as well (elsewhere?), as you clearly include questions of this sort in your data collection.

The authors are grateful for the reviewer's interest in improving the study. However, as indicated in the introduction, values are the attributes that people assign to the landscape, based on their personal perceptions and interpretations. Thus, the concept of landscape value implies the concepts of "perception" and "preference".

Point E5.    You are tackling a very complex type of landscape, which could be characterized as natural, as tourist, as marina, etc. and this makes it a bit more difficult to justify and interpret your findings--which you do very well, nonetheless. However, you should take more into consideration in your discussion that this study involves mostly Spanish visitors and area residents and that their landscape values are not just influenced by personal behaviors (you probably mean personal practices? l. 611), but by a series of factors (perceptions, preferences, norms, values, vested interests etc.), some of which you mention further down, but which are key to the interpretation of the findings--a complex and challenging task, anyway. For instance, concerning the survey respondents' educational level (ls. 494-498), it could be that those with a higher education have more refined, specific or high-standard preferences for beauty and nature (scenic and recreational values) in a landscape than those with a lower education, as marinas are highly artificial environments. You acknowledge the artificiality of the marina later on (top of p.15), but the fact that people may increasingly prefer outdoor/ natural environments does not imply that they will also opt for marinas (ls. 527-533: this whole paragraph seems a bit too speculative). It is indeed a complex question you are coming up against, namely "how artificial do the users of the marina consider its landscape"? The answer, of course, which may help you in the interpretation of your results, depends on a great variety of factors: cultural (i.e. a mostly 'urban culture' or not?); geographical proximity of alternative nature destinations; traditions; dependence of area's livelihoods on the marina, etc., besides, of course, the marina's scenic and recreational values. Just food for thought.

We appreciate all reviewer comments. We have rewritten part of the discussion.

Lines 441-445 (section 4.2) now read: Knowing the respondents' profiles was essential to extract conclusions on how the landscape is perceived. Landscape values ground on how people perceive the landscape and the individual and social attributes they assign to it. In this way, most of the respondents were Spanish and area residents. Thus, the values assigned were biased by these factors, avoiding a richer assessment if a broader sample (foreigners and visitors from other municipalities) were considered.”.

Lines 499-506 (section 4.2) now read: Concerning the educational level, there is a tendency among respondents (not a generalized rule) that the ratings are lower in those with higher education compared to elementary studies. The landscape is formed by the physical elements and the interpretation and transformation the observers make of this reality according to their own experiences [12]. It could be that people with higher education have more refined, specific, or high-level preferences for beauty and nature. Their level of demand is higher when it comes to value. However, only with the data on the education level, it was impossible to draw more conclusions”.

Lines 534--542 (section 4.3) now read: Marinas, as infrastructures on the coast, make the most of the existing natural conditions, inserting themselves without renouncing their artificial nature but maintaining the prominence of the coast [12]. In this sense, marinas should provide additional values that enhance those natural existing in a coastal area. Moreover, a study developed by the United Nations Conference on the impacts of COVID-19 projected growth in preferences for outdoor experiences and contact with nature and water [68]. It supposes a competitive advantage for the marinas landscape. Firstly, these facilities are foundations for the perception of the coastal landscape. Secondly, they enhance existing landscape values by adding new ones”.

Lines 595--597 (section 4.3) now read: The results reaffirm the need to consider landscape as essential for marina management. However, it would be necessary to determine the level of need for nature/artificiality at which the landscape is considered valuable”.

Point E6.    Please, present Appendix A in the text (ls. 180-190) in a clearer way, step by step, in line with your 2-step data collection scheme (ls.164-165).

The reviewers appreciate the comment made by the reviewer, which improves the understanding of the study. The paragraph has been rewritten to make it more understandable.

The text now reads (section 2.2.1): “We used one-to-one semi-structured interviews, which allowed the interviewees free to explain within flexible conduction and tailored to them and their responses to previous questions, taking into account the general aims of the research. In this sense, the interview was prior prepared considering questions grouped in three blocks (see Appendix A). The first one dealt with marina management. The interviewees were asked their opinion on the marina's management, identifying strengths and weaknesses. They were also proposed to provide solutions. The second part examined the idea of landscape. The participants were asked about what they understood as landscape and if they knew ELC. Later, they were asked about how they should improve the landscape within the marina, taking into account the previous ideas carried out in the above topic. The last block looked into public participation by asking in which way and to what extent the public should participate in the decision-making process”.

Point E7.    Specify what you mean by 'high rates' (of what?) both in Table 2 and in the text (l. 583).

We meant to say the amount of money that is charged for goods or services.

Point E8.    Insert 'Threats' in Table 2, in place of 'Weaknesses' which appears twice.

We would like to thank the reviewer for the detail appreciated. We have corrected the error detected by the reviewer.

Point E9.    You may consider ending the Conclusions not only with the positive repercussions of 'increased globalization' but also with a reference to its more negative ones, in light of the broader current quest for more sustainable futures.

We would like to thank the reviewer for the detail appreciated. The conclusions have been rewritten.

Conclusions now reads (section 5): “This study aimed to apply the landscape value framework to evaluate a marina. We provided a set of landscape values obtained from previous SWOT analysis and assessed through a questionnaire conducted with port users. The suitability of the values was analyzed using ANOVA and PCA methods. A survey was carried out, collecting outcomes from 104 users and visitors.

“From the analysis of the survey results, the ANOVA method determined that grouping the items into categories was impossible. It was necessary to deal with the values individually. Likewise, the PCA method established that it was possible to make other different groupings and even reduce the number of items.

“Considering the data collection, first, the SWOT analysis enabled us to collect in an orderly and detailed way the perception of management by the stakeholders. The primary outcomes were that the marina had a privileged location, but there was a lack of space and an excess of urbanization in the surroundings. Marina should be in keeping with an atmosphere of tranquillity and well-being. Interviewees also pointed out the need for coordination between agents related to management. This analysis should include more than just the mere collection of information. The comparison with the management of the marina is closely related to the expression of the degree of satisfaction about it, in addition to representing the reflection of the specific needs of this group.

“Second, the survey gathered the marina’s landscape values. Looking at the overall sample, we found that respondents valued the natural environment, beauty, and sports activities. Nevertheless, there was a need to improve values related to nautical tourism, such as hospitality and maintenance. Other outcomes from the survey were consistent, corroborating the above related and including high rates and excessive urbanization in the surroundings. Finally, there was a common opinion that the landscape is essential to the marina. Marina managers should consider these outcomes and analyze the points of improvement to establish the causes of these disagreements and propose solutions per the established management model.

“Understanding people's motivations and expectations, as well as stakeholder opinions and criteria, are of pivotal importance. The success of achieving a landscape in a marina depends on a delicate interplay of contents and links. Accordingly, the decision-making processes on landscape need to take a multi-scale approach. Such an approach needs to encompass the qualities of the landscape that are important to the in-group (manager and stakeholders), distinguishing them from the out-group and thereby uniting people at this level. Their perception can enable more consensual policies with greater acceptance and involvement. Knowledge of weaknesses and threats may align marina managers with user needs more.

“This study also demonstrates that public participation is helpful to landscape management in marinas. Public participation is a way to articulate people's preferences, considering them with a genuine interest in managing processes. The study proposes a methodology to capture this contribution. Moreover, it shows that management is a delicate interplay between the managers’ desires and the stakeholders and how users perceive this landscape. Ultimately, the landscape should be about investing in local values. Landscape in marinas must be distinct from the global versus local discourse. Increased globalization leads to standards in the quality of services, which creates new scales of abstraction that help people relate to the local features. Furthermore, it has shown the potential to make local landscapes obsolete when activities and products are no longer competitive in the global market”.

The authors reiterate their gratitude for the comments made by the reviewer, as they have improved the clarity and understanding of the manuscript.

This manuscript is a resubmission of an earlier submission. The following is a list of the peer review reports and author responses from that submission.

Round 1

Reviewer 1 Report

Reviewer’s comments:

I have some comments on your paper titled “Landscape values in a marina in Granada (Spain): Enhancing public participation in landscape management”. This study is still weak because it offers little to the scientific literature (e.g., methodology), landscape management, etc.

The following are some my suggestions for how the author(s) could make the paper better:

1. The authors created the Likert questions and employ PCA to analyze the data. The authors should build a PCA-based regression model to identify the factors influencing respondents’ landscape pleasure. In this way, you could identify the most significant values and rank them.

2. In the Discussion Section, you should elaborate on the importance of landscape values and how they affect the behaviors/satisfaction in the light of values and personal/social norms (Kim and Seock, 2019; ROUVEN DORAN and LARSEN, 2016), mindspongecon (Khuc, Q.V. (2022). It is noted that when a person's core values are matched, he or she is likely to feel pleased, joyful, or motivated to act. In other words, you should relate the landscape’s values to the individual’s core mindset (core values) and then evaluate which landscape value is the most important to consider.

3. Please consider some key references below for your additional revising

Kim, S.H., Seock, Y.K., 2019. The roles of values and social norm on personal norms and pro-environmentally friendly apparel product purchasing behavior: The mediating role of personal norms. J. Retail. Consum. Serv. 51, 83–90. https://doi.org/10.1016/j.jretconser.2019.05.023

ROUVEN DORAN, LARSEN, S., 2016. The Relative Importance of Social and Personal Norms in Explaining Intentions to Choose Eco-Friendly Travel Options. Int. J. Tour. Res. 18, 159–166. https://doi.org/10.1002/jtr.2042

Khuc, Q. V. (2022). Mindspongeconomics. Working Paper Series No. 2022/9. https://doi.org/10.31219/osf.io/hnucr

Author Response

Point A0.    I have some comments on your paper titled “Landscape values in a marina in Granada (Spain): Enhancing public participation in landscape management”. This study is still weak because it offers little to the scientific literature (e.g., methodology), landscape management, etc.

The authors would like to acknowledge the effort and work of the reviewer. Her/his contribution always represents a point of improvement for the study.

This study addresses landscape values in an area that has not been studied (marinas). This is pointed out in section 1.1 “Values landscape”:

“The original typology of landscape values was developed by Brown and Reed [20], who established a set of 13 values (aesthetic, recreation, biodiversity, life-supporting, economic, learning, historical, cultural, future, intrinsic, spiritual, therapeutic, subsistence) as part of a forest planning process. This typology has been adapted and used for different applications, such as public lands [21], country management [22-24], urban areas [25-28], rural landscapes [29,30], and coastal landscapes [31,32]. However, there is not a one-to-one correspondence when dealing with places and values [26].

Regarding marina, some scholarships analyze users' perceptions as valuable tools to verify users' satisfaction levels. Yachters' perceptions are used to identify market segmentation [4] or investigate their quality perceptions' effects on their satisfaction with marina services [5]. However, there needs to be more in assessing perceptions and values of the landscape in marinas.”

It also provides a methodology for the selection of values to be considered, as well as procedures for establishing the suitability of the values considered. These goals are represented in the research questions (RQ).

Point A1.    The authors created the Likert questions and employ PCA to analyze the data. The authors should build a PCA-based regression model to identify the factors influencing respondents’ landscape pleasure. In this way, you could identify the most significant values and rank them.

The authors wish to thank the reviewer for her/his observation. As discussed, it is tested whether it was possible to perform the analysis on the categories and not on the totality of the individual topics. But when ANOVA analysis was performed, there was no probability of differences between values in some of the categories. Therefore, this option was discarded.

A factor analysis was performed through PCA. This is a data reduction technique used to find homogeneous groups of variables from a set of numerous variables. In this way, Table 6 provides the results that express the possibility of reducing to 6 the categories to be considered, including the values to be included in each of the categories. The rotation process seeks that the variables saturate a single factor and that the factors contain a reduced number of variables that saturate, if possible, in a single factor. The factorization of each variable shows its importance in each of the new factors considered. Therefore, the importance of each variable is reflected in this table. Moreover, appendix B shows the coefficient matrix of component scores using the regression method has been incorporated.

The text now reads (section 3.2, page 11): We also analyzed the appropriateness of considering the items sets as a single category. The ANOVA model determined that the significant categories were environmental quality (4.22) and aesthetic (3.92). These obtained a high rating, with significant internal consistency. On the contrary, it was found that certain groups needed to have adequate reliability because they had Cronbach's alpha values lower than 0.7: “Learning” (0.689), “Recreation” (0.649), and “Future” (0.471). Nevertheless, groups with high F value and low significance (Sig. < 0.05) means that there were likely to be differences between the values that comprised it. Therefore, the items should be analyzed individually rather than as group members. Categories with different values or even reducing some initially considered values would be necessary. These results were consistent with outcomes from PCA. Table 6 shows the results of the component matrix after rotation. Values under 0.5 were excluded (for PCA-based regression model see appendix B).”

Point A2.    In the Discussion Section, you should elaborate on the importance of landscape values and how they affect the behaviors/satisfaction in the light of values and personal/social norms (Kim and Seock, 2019; Rouven Doran and Laresen, 2016), mindspongecon (Khuc, Q.V. (2022). It is noted that when a person's core values are matched, he or she is likely to feel pleased, joyful, or motivated to act. In other words, you should relate the landscape’s values to the individual’s core mindset (core values) and then evaluate which landscape value is the most important to consider.

The authors are grateful for the approach offered by the reviewer. The articles indicated are interesting. Personal norms influence people's behavior and condition the way they act in relation to their environment. In this sense, values, understood as the principles that govern behavior, play a significant role in explaining certain beliefs, attitudes and behaviors. However, the subject of this study is landscape values, i.e., the attributes that people assign to the perceived landscape.

It is evident that the perception of the environment is conditioned by personal and social norms. As it is pointed out by the reviewer, it may be interesting to study these norms in the assignment of value to a landscape. In this sense, this option is suggested as future research.

The text now reads (section 4.4, page 17): Personal behaviors influence landscape values. Personal norms are internal personal obligations that condition behavior [76,77]. People's basic mentality (values) and, therefore, their behavior condition how they perceive their environment. In this sense, it is possible to select landscape values concerning the importance for people based on their personal norms”.

Point A3.     Please consider some key references below for your additional revising.

Kim, S.H., Seock, Y.K. (2019). The roles of values and social norm on personal norms and pro-environmentally friendly apparel product purchasing behavior: The mediating role of personal norms. J. Retail. Consum. Serv. 51, 83–90. https://doi.org/10.1016/j.jretconser.2019.05.023

Rouven, D., Larsen, S. (2016). The Relative Importance of Social and Personal Norms in Explaining Intentions to Choose Eco-Friendly Travel Options. Int. J. Tour. Res. 18, 159–166. https://doi.org/10.1002/jtr.2042

Khuc, Q.V. (2022). Mindspongeconomics. Working Paper Series No. 2022/9. https://doi.org/10.31219/osf.io/hnucr

We appreciate the reviewer's contribution. In addition, from future research relating the most appropriate landscape values to personal norms, two of the references provided by the reviewer have been incorporated

[76] Doran, R.; Larsen, S. The Relative Importance of Social and Personal Norms in Explaining Intentions to Choose Eco-Friendly Travel Options. Int. J. Tour. Res. 2016 18, 159–166. https://doi.org/10.1002/jtr.2042.

[77] Kim, S.H.; Seock, Y.K. The roles of values and social norm on personal norms and pro-environmentally friendly apparel product purchasing behavior: The mediating role of personal norms. J. Retail. Consum. Serv. 2019 51, 83–90. https://doi.org/10.1016/j.jretconser.2019.05.023.

The authors reiterate their gratitude for the comments made by the reviewer, as they have improved the clarity and understanding of the manuscript

Reviewer 2 Report

"This study demonstrates that public participation is helpful in landscape management in marinas. " the value and purpose of this manuscript is not "public participation". Finding a way or a process to improve the marina’s landscape values through  "public participation". That is the true value of this study. 

The paper presents some interesting contextual data, but it simply reconfirms existing knowledge and relations in a new context. The authors need to revise from a target that is not with the  "public participation".

Moreover, some highlights are covered in this manuscript. It will give a poor score from the reviewers

"The European Landscape Convention (ELC) encourage people's involvement and promotes their participation in landscape assessment and planning. " For research articles, abstracts should give a pertinent overview of the work. It is essential to place the question in a broad context and highlight the purpose of the study. The encouragement of ELC is essential. But the view of the article needs to be further than a  Convention. That is the value and purpose of this manuscript. 

From "2. Materials and Methods",  the methods were not clearly shown. "Principal component analysis (PCA)" and "The analysis of variance (ANOVA)" are both methods, they are not suitable in "2.3. Data analysis". "Survey" is not a scientific article method. A way like ANOVA will be more suitable.

An interview sample should be shown in Appendix A.

The conclusions also need to be revised with those comments.

Author Response

Point B0.    "This study demonstrates that public participation is helpful in landscape management in marinas" the value and purpose of this manuscript is not "public participation". Finding a way or a process to improve the marina’s landscape values through "public participation". That is the true value of this study.

The paper presents some interesting contextual data, but it simply reconfirms existing knowledge and relations in a new context. The authors need to revise from a target that is not with the "public participation".

Moreover, some highlights are covered in this manuscript. It will give a poor score from the reviewers.

The authors would like to thank the reviewer for his dedication and time. The purpose of the study, as reflected in the research questions (RQ), is to determine the values of the landscape. Collaterally, the procedure followed to perform the valuation also represents a form of public participation in the landscape. This is therefore a secondary issue, but the results may be of interest for the management of the marina.

Point B1.    The European Landscape Convention (ELC) encourage people's involvement and promotes their participation in landscape assessment and planning. " For research articles, abstracts should give a pertinent overview of the work. It is essential to place the question in a broad context and highlight the purpose of the study. The encouragement of ELC is essential. But the view of the article needs to be further than a Convention. That is the value and purpose of this manuscript.

We would like to thank the reviewer for the detail appreciated. The basis on which the summary was written is accurate and clear. In line with what has been stated in the previous point, the summary reinforces the issue of landscape values, reducing public participation to a collateral issue.

The abstract now reads (page 1): This study focused on identifying and assessing a marina’s landscape values. We took Marina del Este (Granada, Spain) as a case study. We considered interviews and a general survey to enhance the participation of stakeholders and the public, embracing different sensibilities and standpoints. First, the SWOT analysis from interviews with stakeholders enabled us to collect management's perceptions. Second, the survey gathered the marina's landscape values, comprising 104 respondents from visitors and users. ANOVA and PCA methods were applied to check suitability of the values. The results showed marina should be in keeping with an atmosphere of tranquillity and well-being. Nevertheless, there was a need to improve values related to nautical tourism, such as hospitality and maintenance, dealing with the lack of space and an excess of urbanization in the surroundings. Marina managers should consider these outcomes and analyze the points of improvement to establish the causes of these disagreements and propose solutions for the established management model. The perception of stakeholders and users can enable more consensual policies with greater acceptance and involvement. This study also demonstrates that public participation is helpful in landscape management in marinas”.

Point B2.    From "2. Materials and Methods", the methods were not clearly shown. "Principal component analysis (PCA)" and "The analysis of variance (ANOVA)" are both methods, they are not suitable in "2.3. Data analysis". "Survey" is not a scientific article method. A way like ANOVA will be more suitable.

The authors would like to thank the reviewer for his detailed and precise comments. In section 2.2, "survey" has been changed to "questionnaire", which is more appropriate. The title of section 2.3 has also been changed to adapt it to the content of the methods used in the analysis of the results. A short text has also been added to facilitate the development of the procedure.

The text now reads (section 2.3, page 7): “2.3. Data analysis procedure. The suitability of the values was analyzed using two main methods. First, reliability of the outcomes was determined though Cronbach’s alpha. Second, ANOVA method was applied to check whether it was appropriate to group the items in the categories regarded, considering the value of the category as the average value of the topics that integrate it. Finally, PCA method was used to evaluate which items could be narrowed down, as well as to identify the most influential factors. For data processing and analysis, we used SPSS©.”

Point B3.    An interview sample should be shown in Appendix A.

We appreciate the review`s advice. As discussed in section 2.2.1 (page 8), the interview featured three parts in which participant were allowed to freely express their opinions. It was more of a guided conversation than an interview with sample questions. This is why it is not reflected in any appendix.

Point B4.    The conclusions also need to be revised with those comments.

The comment made by the reviewer is entirely appropriate. The conclusion needs to be rewritten taking into account the focus on landscape values.

The conclusions now read (section 5, page 18): “This study aimed to apply the landscape value framework to evaluate a marina. We provided a set of landscape values obtained from previous SWOT analysis and assessed through a questionnaire conducted with port users. The suitability of the values was analyzed using ANOVA and PCA methods. A survey was carried out, collecting outcomes from 104 users and visitors.

“From the analysis of the survey results, the ANOVA method determined that grouping the items into categories was impossible. It was necessary to deal with the values individually. Likewise, the PCA method established that it was possible to make other different groupings and even reduce the number of items.

“Considering the study design, first, the SWOT analysis enabled us to collect in an orderly and detailed way the perception of management by the stakeholders. The primary outcomes were that the marina had a privileged location, but there was a lack of space and an excess of urbanization in the surroundings. Marina should be in keeping with an atmosphere of tranquillity and well-being. Interviewees also pointed out the need for coordination between agents related to management. This analysis should include more than just the mere collection of information. The comparison with the management of the marina is closely related to the expression of the degree of satisfaction about it, in addition to representing the reflection of the specific needs of this group.

“Second, the survey gathered the marina’s landscape values. Looking at the overall sample, we found that respondents valued the natural environment, beauty, and sports activities. Nevertheless, there was a need to improve values related to nautical tourism, such as hospitality and maintenance. Other outcomes from the survey were consistent, corroborating the above related and including high rates and excessive urbanization in the surroundings. Finally, there was a common opinion that the landscape is essential to the marina. Marina managers should consider these outcomes and analyze the points of improvement to establish the causes of these disagreements and propose solutions per the established management model.

“Understanding people's motivations and expectations, as well as stakeholder opinions and criteria, are of pivotal importance. The success of achieving a landscape in a marina depends on a delicate interplay of contents and links. Accordingly, the decision-making processes on landscape need to take a multi-scale approach. Such an approach needs to encompass the qualities of the landscape that are important to the in-group (manager and stakeholders), distinguishing them from the out-group and thereby uniting people at this level. Their perception can enable more consensual policies with greater acceptance and involvement. Knowledge of weaknesses and threats may align marina managers with user needs more.

“This study also demonstrates that public participation is helpful to landscape management in marinas. Public participation is a way to articulate people's preferences, considering them with a genuine interest in managing processes. The study proposes a methodology to capture this contribution. Moreover, it shows that management is a delicate interplay between the managers’ desires and the stakeholders and how users perceive this landscape. Ultimately, the landscape should be about investing in local values. Landscape in marinas must be distinct from the global versus local discourse. Increased globalization leads to standards in the quality of services, which creates new scales of abstraction that help people relate to the local features. Furthermore, it has shown the potential to make local landscapes obsolete when activities and products are no longer competitive in the global market.”

The authors reiterate their gratitude for the comments made by the reviewer, as they have improved the clarity and understanding of the manuscript.

Round 2

Reviewer 1 Report

Thank you for carefully and appripriately addressing most of my comments. The revised manuscript is much improved and I have no further comments on your work.

Author Response

The authors would like to thank the reviewer for the indications that have allowed us to improve the manuscript.

We are also grateful for the interest shown and the dedication taken in the review

Reviewer 2 Report

Unfortunately, the authors did not respond to the reviewers' comments one by one.

Author Response

Point B1.    "This study demonstrates that public participation is helpful in landscape management in marinas" the value and purpose of this manuscript is not "public participation". Finding a way or a process to improve the marina’s landscape values through "public participation". That is the true value of this study.

The authors would like to thank the reviewer for his dedication and time. The purpose of the study, as reflected in the research questions (RQ), is to determine the values of the landscape. It proposes a methodology to collect data and procedures to verify the accuracy of the outcomes. Collaterally, the procedure followed to perform the valuation also represents a form of public participation in the landscape. This is therefore a secondary issue, but the results may be of interest for the management of the marina. In this way, the abstract and the introduction have been rewritten.

The study objectives now read (section 1.3, page 2): “This study seeks to apply the landscape value framework to the evaluation of a marina. The limited landscape research in marinas is why the landscape values in these maritime facilities have yet to be studied. It also represents an opportunity to enhance public participation in marinas' landscape management, which provides outcomes from a theoretical framework to improve management. Public participation should include experts and people, but it also must recognize the different stakeholders and social groups [41,42]. Attending to Eitier and Vik [43] and translated to marinas, management implies a mutual commitment between parties: managers, stakeholders, and users. Moreover, managers should enhance the participation of the public and other relevant stakeholders in the landscape policies [44]. We addressed the following question:

“RQ1.   Which are the landscape values identified, and how users perceived the marina landscape?

“RQ2.   How is it possible to verify the suitability of the chosen values?

“RQ3.   How could the landscape value framework enhance public participation in landscape management?

Knowing the links between people and the marina is the primary goal of marina managers. Landscape values reflect the marina user perceptions. It represents a feedback tool through which marina managers can verify whether the policies implemented have a satisfactory response from users. It also represents a set of expectations to satisfice. Marina managers must learn from customers` needs and anticipate future expectations [14]. It also verifies whether the strategic planning developed aligns with what the stakeholders perceive.”

Point B2.    The paper presents some interesting contextual data, but it simply reconfirms existing knowledge and relations in a new context. The authors need to revise from a target that is not with the "public participation".

Moreover, some highlights are covered in this manuscript. It will give a poor score from the reviewers.

We appreciate all reviewer comments that can be used to improve the study. As discussed in section 1.1, landscape values have been applied in several areas, but not in marinas. The novelty of this study represents the study of landscape values in an area where no previous work has been done.

As discussed in the previous point, public participation is a collateral element. It represents a tool through which the study has been carried out.

Point B3.    The European Landscape Convention (ELC) encourage people's involvement and promotes their participation in landscape assessment and planning. " For research articles, abstracts should give a pertinent overview of the work. It is essential to place the question in a broad context and highlight the purpose of the study. The encouragement of ELC is essential. But the view of the article needs to be further than a Convention. That is the value and purpose of this manuscript.

We would like to thank the reviewer for the detail appreciated. The basis on which the summary was written is accurate and clear. In line with what has been stated in the previous point, the summary reinforces the issue of landscape values, reducing public participation to a collateral issue.

The abstract now reads (page 1): Landscape values are related to the attributes that people assign to perceived landscape. They reflect marina user perceptions, which represents a feedback tool for marina managers to verify the degree of user satisfactory. This study focused on identifying and assessing a marina’s landscape values. We took Marina del Este (Granada, Spain) as a case study. We considered interviews and a questionnaire to enhance the participation of stakeholders and users. First, the SWOT analysis from interviews with stakeholders enabled us to collect management's perceptions. Second, the survey gathered the marina's landscape values, comprising 104 respondents from visitors and users. ANOVA and PCA methods were applied to check suitability of the values. The results showed marina should be in keeping with an atmosphere of tranquillity and well-being. Nevertheless, there was a need to improve values related to nautical tourism, such as hospitality and maintenance, dealing with the lack of space and an excess of urbanization in the surroundings. Marina managers should consider these outcomes and analyze the points of improvement to establish the causes of these disagreements and propose solutions for the established management model. The perception of stakeholders and users can enable more consensual policies with greater acceptance and involvement”.

Point B4.    From "2. Materials and Methods", the methods were not clearly shown. "Principal component analysis (PCA)" and "The analysis of variance (ANOVA)" are both methods, they are not suitable in "2.3. Data analysis". "Survey" is not a scientific article method. A way like ANOVA will be more suitable.

The authors would like to thank the reviewer for his detailed and precise comments. To improve the understanding of the steps followed in the methodology, the name of section 2.2 has been changed to "Data collection". This section includes subsections 2.2.1 "Interviews" and 2.2.2 "Questionnaire, which is more appropriate than “Survey” Title of section 2.3 has also been changed to adapt it to the content of the methods used in the analysis of the results. A short text has also been added to facilitate the development of the procedure.

The text now reads (section 2.3, page 7): “2.3. Data analysis procedure. The suitability of the values was analyzed using two main methods. First, reliability of the outcomes was determined though Cronbach’s alpha. Second, ANOVA method was applied to check whether it was appropriate to group the items in the categories regarded, considering the value of the category as the average value of the topics that integrate it. Finally, PCA method was used to evaluate which items could be narrowed down, as well as to identify the most influential factors. For data processing and analysis, we used SPSS©.”

Point B5.    An interview sample should be shown in Appendix A.

We appreciate the review`s advice. A new appendix A has been added that includes the example of the questions that were asked to the interviewees, as well as table 2, which contained the marina landscape values’ definition.

Point B6.    The conclusions also need to be revised with those comments.

The comment made by the reviewer is entirely appropriate. The conclusion needs to be rewritten taking into account the focus on landscape values.

The conclusions now read (section 5, page 18): “This study aimed to apply the landscape value framework to evaluate a marina. We provided a set of landscape values obtained from previous SWOT analysis and assessed through a questionnaire conducted with port users. The suitability of the values was analyzed using ANOVA and PCA methods. A survey was carried out, collecting outcomes from 104 users and visitors.

“From the analysis of the survey results, the ANOVA method determined that grouping the items into categories was impossible. It was necessary to deal with the values individually. Likewise, the PCA method established that it was possible to make other different groupings and even reduce the number of items.

“Considering the study design, first, the SWOT analysis enabled us to collect in an orderly and detailed way the perception of management by the stakeholders. The primary outcomes were that the marina had a privileged location, but there was a lack of space and an excess of urbanization in the surroundings. Marina should be in keeping with an atmosphere of tranquillity and well-being. Interviewees also pointed out the need for coordination between agents related to management. This analysis should include more than just the mere collection of information. The comparison with the management of the marina is closely related to the expression of the degree of satisfaction about it, in addition to representing the reflection of the specific needs of this group.

“Second, the survey gathered the marina’s landscape values. Looking at the overall sample, we found that respondents valued the natural environment, beauty, and sports activities. Nevertheless, there was a need to improve values related to nautical tourism, such as hospitality and maintenance. Other outcomes from the survey were consistent, corroborating the above related and including high rates and excessive urbanization in the surroundings. Finally, there was a common opinion that the landscape is essential to the marina. Marina managers should consider these outcomes and analyze the points of improvement to establish the causes of these disagreements and propose solutions per the established management model.

“Understanding people's motivations and expectations, as well as stakeholder opinions and criteria, are of pivotal importance. The success of achieving a landscape in a marina depends on a delicate interplay of contents and links. Accordingly, the decision-making processes on landscape need to take a multi-scale approach. Such an approach needs to encompass the qualities of the landscape that are important to the in-group (manager and stakeholders), distinguishing them from the out-group and thereby uniting people at this level. Their perception can enable more consensual policies with greater acceptance and involvement. Knowledge of weaknesses and threats may align marina managers with user needs more.

“This study also demonstrates that public participation is helpful to landscape management in marinas. Public participation is a way to articulate people's preferences, considering them with a genuine interest in managing processes. The study proposes a methodology to capture this contribution. Moreover, it shows that management is a delicate interplay between the managers’ desires and the stakeholders and how users perceive this landscape. Ultimately, the landscape should be about investing in local values. Landscape in marinas must be distinct from the global versus local discourse. Increased globalization leads to standards in the quality of services, which creates new scales of abstraction that help people relate to the local features. Furthermore, it has shown the potential to make local landscapes obsolete when activities and products are no longer competitive in the global market.”

The authors reiterate their gratitude for the comments made by the reviewer, as they have improved the clarity and understanding of the manuscript.
